# Distributed context-dependent choice information in mouse posterior cortex

Javier G. Orlandi[1,3], Mohammad Abdolrahmani[1], Ryo Aoki[1], Dmitry R. Lyamzin [1] &
Andrea Benucci [1,2] ✉

Choice information appears in multi-area brain networks mixed with sensory, motor, and cognitive variables. In the posterior cortex—traditionally implicated in decision computations—the presence, strength, and area specificity of choice signals are highly variable, limiting a cohesive understanding of their computational significance. Examining the mesoscale activity in the mouse posterior cortex during a visual task, we found that choice signals defined a decision variable in a low-dimensional embedding space with a prominent contribution along the ventral visual stream. Their subspace was near-orthogonal to concurrently represented sensory and motor-related activations, with modulations by task difficulty and by the animals' attention state. A recurrent neural network trained with animals' choices revealed an equivalent decision variable whose context-dependent dynamics agreed with that of the neural data. Our results demonstrated an independent, multi-area decision variable in the posterior cortex, controlled by task features and cognitive demands, possibly linked to contextual inference computations in dynamic animal–environment interactions.

The view of discrete neural modules in cortical networks that selectively encode sensory, decisional, or motor processes[1,2], has been challenged by the evidence of mixed representations within and across neurons[3]. In the context of decision-making computations, mixed selectivity reflects complex interactions of task and behavioral variables with decision information[4–6], with the prominence of decision signals being influenced by a diversity of components, such as the stimulus-coding strength of neurons[7,8], the correlation properties of the network[9–11], modulatory signals associated with changeable states of attention[12], which can also contextually enable, route, and gate decision-related information[13]. The area location along the sensory hierarchy[14,15] and even the strategy used by an animal to solve a task can affect the detectability of decision signals in neural circuits[16].

Together, these observations have underscored the difficulty to identify decision signals and separate them from co-represented perceptual, motor, and cognitive variables. This challenge has not been unique to primate studies, with the mouse animal model playing an increasingly prominent role in decision-making studies[17] in view of the abundant tools available for the dissection of neural circuits[18]. In this species, possibly more prominently than in larger mammals, task-instructed and uninstructed movement-related activations have been observed with large amplitude even in early sensory regions[19–22]. This has further challenged the separation of decision from movement variables, particularly in tasks with freely moving mice or in virtual-reality navigation tasks in which motor signals continuously affect cortex-wide networks, with head and body orienting movements predictive of choice[23] (see Supplementary Table 1 for a summary of related studies). In consideration of this complex superposition of variables with decision signals, some studies have attempted to isolate decision components by minimizing other possibly co-represented signals (e.g., short-term memory, novelty, navigation, evidence accumulation processes during visually guided behavior). Notably, these studies (hereafter, visually guided tasks, for brevity) could not detect significant choice information in posterior sensory and associative

[1]RIKEN Center for Brain Science, 2-1 Hirosawa, Wako-shi, Saitama 351-0198, Japan. [2]University of Tokyo, Graduate School of Information Science and Technology, Department of Mathematical Informatics, 1-1-1 Yayoi, Bunkyo City, Tokyo 113-0032, Japan. [3]Present address: Department of Physics and Astronomy, University of Calgary, Calgary, Alberta T2N 1N4, Canada. ✉e-mail: andrea.benucci@riken.jp

cortices[24–27], representing a departure from primate studies which instead could detect choice information even in early visual areas during similar tasks[7,28–34].

Here, we sought to identify signatures of choice information across multiple areas in the mouse posterior cortex during a visually guided task, examining the cortical-area specificity and representational dependencies of choice signals with other variables co-activating these networks. To this end, we introduced two novel elements in our experimental and analytical design: first, we trained animals in a complex variant of an orientation discrimination task[17], aiming to maximize cognitive demands based on perceptual information, but without introducing memory, novelty, navigation, or evidence accumulation components. Second, we applied a tensor decomposition method[35] combined with activity-mode analysis[36] on mesoscale recordings of the posterior cortex (imaging of GCaMP signals); this analysis enabled the detection of signals even if sparsely represented among neurons and distributed across broad regions irrespective of classic area boundaries, such as those defined by retinotopic mapping[35].

## Results

### Mesoscale imaging of the posterior cortex during a discrimination task

Using an automated setup featuring voluntary fixation of the animals' heads[37] (Fig. 1a), we trained mice (n = 7) to carry out a complex version of a two-alternative forced choice (2AFC) orientation discrimination task[17]. The animals had to use their front paws to rotate a toy wheel[38] that controlled the horizontal position of two circular grating stimuli presented on a screen positioned in front of them. Each stimulus was presented at monocular eccentricities with orientations that varied from trial to trial. To obtain a water reward, mice had to shift the stimulus most similar to a learned target orientation to the center of the screen (Fig. 1b, c), with the actual target orientation rarely shown to the animal. Therefore, difficulty had an invariance to absolute orientations, which had to be ignored by the animal and depended only on the relative orientation between the two stimuli[17]. After animals reached performance levels above 75% correct (Fig. 1d), we used a macroscope to image mesoscale GCaMP responses in 10 posterior cortical areas (Fig. 1e; Methods). In individual trials, the neural activity was highly variable, with response activity associated with the onset of visual stimuli and movements of the limbs, trunk, and eyes, as recently described[21] (Fig. 1f).

### Decomposition of neural responses

To extract different variables from the neural signal and map them onto defined cortical regions, we adopted a recent variant of non-negative matrix factorization—locaNMF[35]. This decomposition method identifies tensor components associated with specified seeding regions. When seeding on a given area, locaNMF decomposes the signal into a sum of separable spatial-temporal tensors, with spatial components constrained by the seeding region and temporal components representing the scaling amplitudes of the spatial components. These temporal vectors are potentially more informative than a single vector computed as the average across spatial locations (pixels) within a given area[35]. We aligned all imaging sessions according to the

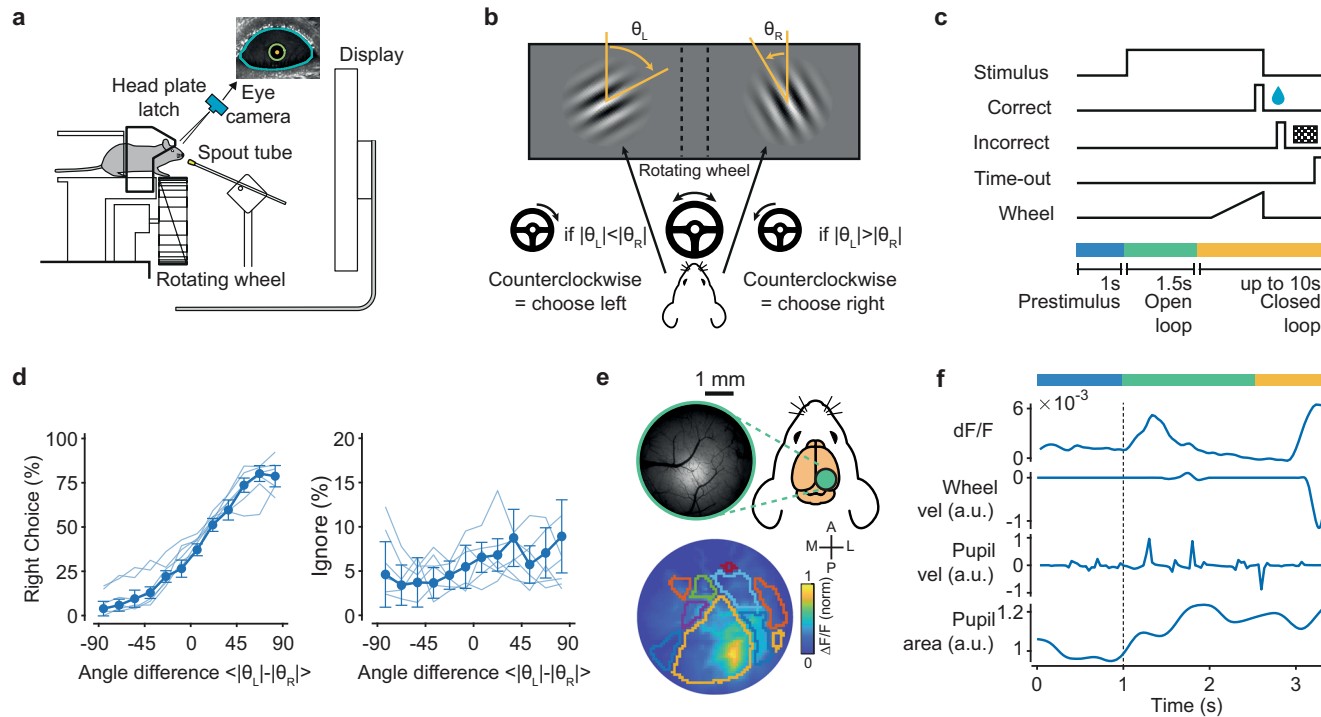

**Fig. 1 | Imaging the posterior cortex during an orientation discrimination task.** **a** Mice were trained on a 2AFC orientation discrimination task using an automated setup featuring voluntary head fixation. They signaled a L/R choice by rotating a toy wheel with their front paws. Schematic adapted from[37]. **b** Mice rotated the wheel to position the most vertical of two oriented gratings in the center of the screen. **c** Trial structure: After a 1 s pre-stimulus period, the stimulus was presented, followed by a 1.5 s open-loop (OL) interval in which wheel movements were decoupled from stimulus movements. Thereafter, in the closed-loop (CL) period, wheel rotations resulted in L/R horizontal shifts of the stimuli. Correct choices were rewarded with water; incorrect choices were followed by a checkerboard pattern presentation. Ten seconds of no movement in the CL period triggered a time-out period. **d** Left: mice's performance in the task (fraction of right choices) as a function of relative angle difference from the target orientation (nominal value of zero), i.e., the task difficulty, averaging across trials with combinations of left and right angles associated to the same difficulty level. Right: fraction of timeout trials as a function of angle difference from the target orientation. Timeout trials did not depend on task difficulty. Thick line = mean (±s.e.) across animals; thin lines = individual animals. (n = 7 animals). **e** Widefield calcium imaging of the posterior cortices of Thy1-GCaMP6f mice, with retinotopic mapping of 10–12 visual areas (colored contours). **f** Simultaneously recorded average fluorescence signal (dF/F), wheel and eye velocities, and pupil area. In this example, choice was signaled at t = 3.1 s (by a sharp increase in wheel velocity). Dashed line represents the stimulus onset time.

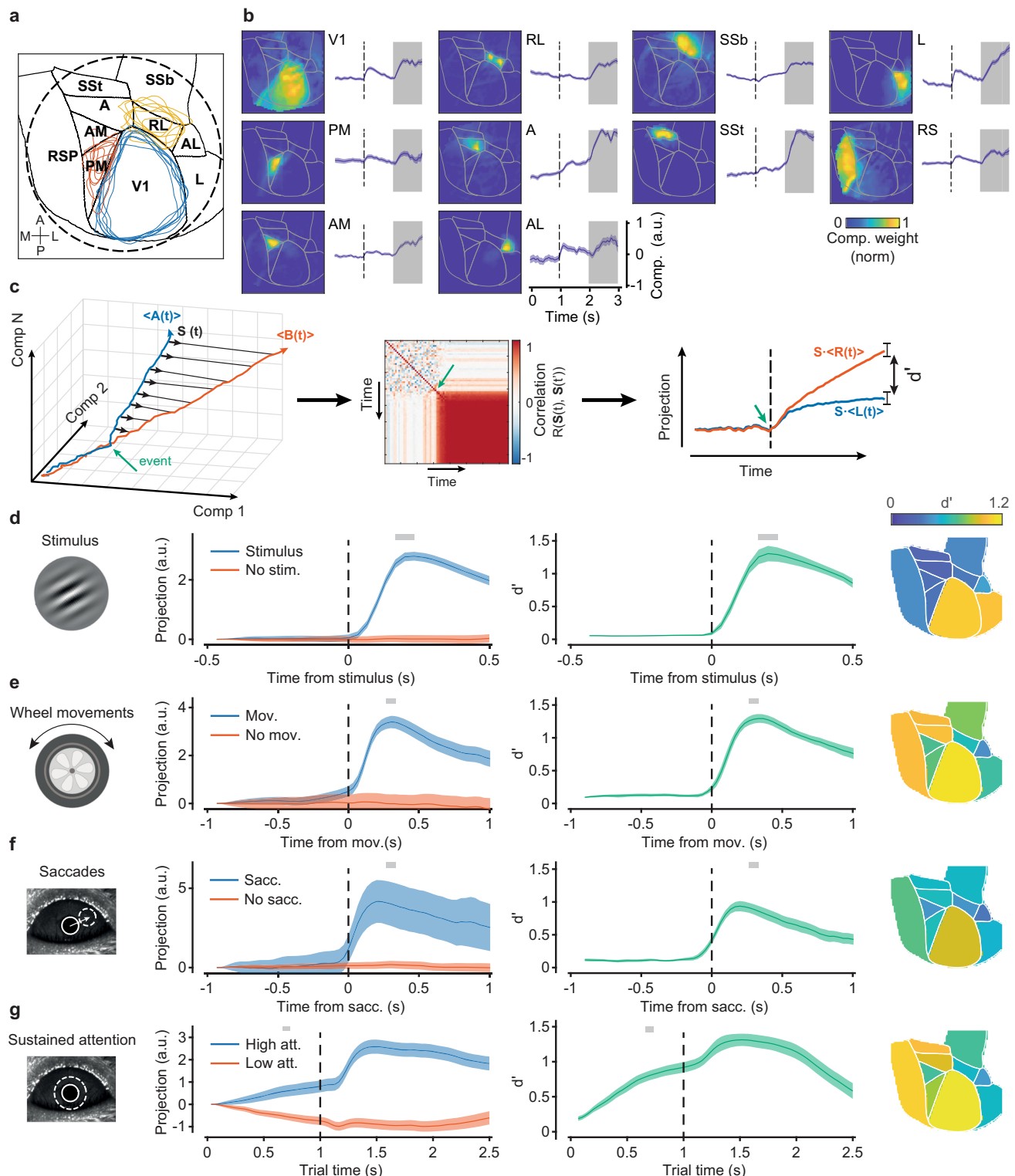

Allen Common Coordinate Framework[39] (Fig. 2a) and seeded the initial spatial decomposition using 10 large regions centered on retinotopically identified areas (based on field sign maps, Supplementary Fig. 1a, see also ref. [20]) that extended significantly beyond area boundaries (Supplementary Fig. 1b). Consistent with the initial seeding, the factorization typically converged toward components with peak amplitudes within individual retinotopic areas (Fig. 2b, see Supplementary

Fig. 2 for an example decomposition). Depending on the seeding region, associated temporal components differentially emphasized sensory or behavioral variables; for instance, when seeding on the primary visual cortex, the largest component (in explained variance, EV) clearly highlighted a stimulus-evoked response (Fig. 2b). The largest components within the parietal regions[40] showed negligible visually driven responses and strong movement-related activations

**Fig. 2 | LocaNMF decomposition identifies sensory, behavioral, and attention-related variables. a** Characteristic imaging window (dashed circle) superimposed on 10 cortical areas from the Allen Brain Atlas reference framework. Blue, red, and yellow contours are the reference-aligned area boundaries for V1, PM and RL for each animal. **b** Spatial weights and trial-averaged time-series of the largest locaNMF components for each of the 10 seeding regions (Supplementary Fig. 1a) for a representative animal. This average consisted of trials with wheel movements within the 1 s shaded time interval, collectively for clock-wise and cc-wise wheel rotations. Dashed lines denote stimulus onset. **c** Schematic for the definition of state axes. The direction of the state axis becomes stable after an event indicated by the green arrow. Vector stability is measured as the temporal autocorrelation R(**S**(t), **S**(t′)), (right panel). Projections (cross-validated) of the two variables **A**(t) and **B**(t) onto **S** separately over time, as quantified by a d′ discriminability measure.

**d** Stimulus-related state axes. Left: projections of trials with and without a stimulus response onto the stimulus state axis for a given animal. Lines and shaded regions indicate across-trial projection averages and its 95% CI. Middle: Discriminability d′ over time, averaged across all animals ($n = 7$, line for mean and shaded area for its s.e.). Gray bars on top, epoch used for the time average of the state axis. Right: area-specific peak d′ scores obtained by defining the state axis using only the components originating from that area averaged across animals. **e** As in **d**, but aligned to movement-detection time, i.e., separability between trials with and without a detected wheel movement. **f**, as in **e**, but for saccadic eye movements. **g** As in **f**, but for sustained attention. Trials with high and low levels of sustained attention were defined based on pupil area changes, using the highest or lowest 33$^{rd}$ percentile of the area-change distribution.

(Fig. 2b). The largest locaNMF component of each cortical area provided significant explanatory power, together contributing to 96% of the total explained variance (Supplementary Fig. 1c). These contributions being highly heterogeneous, with the largest component of V1 accounting for 45% of the total variance. By contrast, the first PCA component contributed, on average, approximately 85% of EV, being strongly influenced by large amplitude movement-related activations[19]. For each area, the number of components significantly contributing to the EV (Methods) was not directly proportional to surface area; for instance, areas AL and L had commensurate surface area and contributed similarly to the overall EV, but L required about twice as many components as AL (Supplementary Fig. 1d–g), in agreement with the different cortical localizations of task and behavioral variables.

To identify these variables in locaNMF components, we defined state axes in a multi-dimensional space of component activations (Fig. 2c). This approach further reduced the dimensionality of the data by isolating activity dimensions that linearly discriminated pairs of variables. For instance, the stimulus axis captured the onset of the visual stimulus, remaining stable after the stimulus' appearance (Supplementary Fig. 3a), and with the projected locaNMF components deviating from the baseline about 200 ms after stimulus onset (Fig. 2d). We quantified the time-dependent increase in the detectability of stimulus components using a d′ discriminability measure, which can be linked to Fisher information[41,42], bounding the variance for estimating a population-encoded parameter. This resulted in d′ values greater than one at the peak of stimulus response (Fig. 2d; $1.38 \pm 0.13$, mean ± standard error, s.e.). Using only the LocaNMF components from a particular seeding region, allowed us to also quantify the relative contribution of that area to the d′ discriminability. For the stimulus variable, the primary and secondary visual cortices (V1, L) had the largest discriminability (d′ = $1.10 \pm 0.09$ and $1.12 \pm 0.13$, respectively), followed by area AL (d′ = $0.51 \pm 0.06$). When attempting to discriminate the orientation of the contralateral visual stimulus, no area carried sufficient information, even for the most dissimilar orientation pairs (Supplementary Fig. 4), as expected from the lack of orientation domains in the mouse visual cortex[43] and the spatial resolution of mesoscale imaging. In control experiments, we used a dual-wavelength imaging approach to quantify the effect of the hemodynamic component[44]. Measurements of the calcium-independent GCaMP fluorescence showed that locaNMF components, state axes, and discriminability values were not significantly affected by the hemodynamic signal (Supplementary Fig. 5).

Besides bottom-up visual inputs, imaged posterior regions reflected activations associated with general movements of the body and eyes[20]. Therefore, we defined state axes associated with wheel and saccadic eye movements. Projections onto these axes resulted in high discriminability of both types of movements (Fig. 2e, f; peak d′ = $1.29 \pm 0.07$ and $0.94 \pm 0.08$ for wheel and eye movements, respectively). Area-specific projections highlighted larger contributions by anterior-medial areas (Fig. 2e, f; Supplementary Fig. 3b, c), with d′

values increasing before or coincidentally with the detection of movements, suggesting pre-motor contributions, e.g., corollary discharges[45], and reaching values greater than one after movement execution. These axes remained stable after event onset, as shown by their cross-correlograms (Supplementary Fig. 3b, c and Supplementary Fig. 6).

We also identified aspects of the variability in locaNMF components that depended on the attention state of the animal. Underlying changes in sustained attention can be both task-related (e.g., engagement or motivational state) and task-independent components (e.g., arousal or alertness)[46,47]. Accordingly, in individual sessions, we observed fluctuations in performance that correlated with changes in pupil dilations and reaction times (Supplementary Fig. 7a, b)—two biomarkers associated with changes in sustained attention[20,48]. Based on the pupil area increase within a short time window after stimulus presentation (open-loop, Methods), we defined a state axis that discriminated between states of high and low sustained attention (Fig. 2g). Associated d′ values deviated significantly from zero largely before stimulus onset (after imposed zero discriminability at trial onset; see Methods). Discriminability values reached d′ = 0.5 approximately 0.5 s after trial onset and remained above this value throughout the trial duration, with peak d′ = $1.31 \pm 0.09$. The state axis defined by attentional modulations remained stable throughout the duration of the trial (Supplementary Fig. 3d), consistent with periods of high and low sustained attention that persisted across trials[20]. The attention state axis was stable relative to the trial outcome (correct or incorrect); the angle between the state axes for sustained attention defined using either correct or incorrect trials was $23° \pm 2°$, slightly larger than the expected value for parallel vectors given the variability in the data, that is, the average angle between the same state axis defined using different folds of the cross-validation procedure (-13° on average, Supplementary Fig. 8c). The d′ values obtained when discriminating attention states from correct trials using their projections onto the state axis defined with incorrect trials and vice versa were comparable (d′ = $1.17 \pm 0.07$ and d′ = $1.2 \pm 0.1$ respectively). Finally, when looking at the spatial contributions to discriminability, the anterior-medial visual areas and the retrosplenial cortex contributed the most to large d′ discriminability (Fig. 2g, Supplementary Fig. 3d).

Together, these results showed that sensory inputs, movement-related activations, and attentional signals were concurrently present in the posterior cortical regions, and could be separated by the locaNMF tensor decomposition, permitting the identification of their characteristic spatial and temporal signatures.

## Choice signals

This approach also allowed us to identify choice-related signals. We adopted an operational definition of choice as signals that correlated with animals' L/R decisions, independently of the stimulus and with premotor signatures reflecting action selection[49]. We considered trials in which the first detected wheel rotation occurred at least half a second after stimulus onset. When occurring in the open-loop (Fig. 1c), the

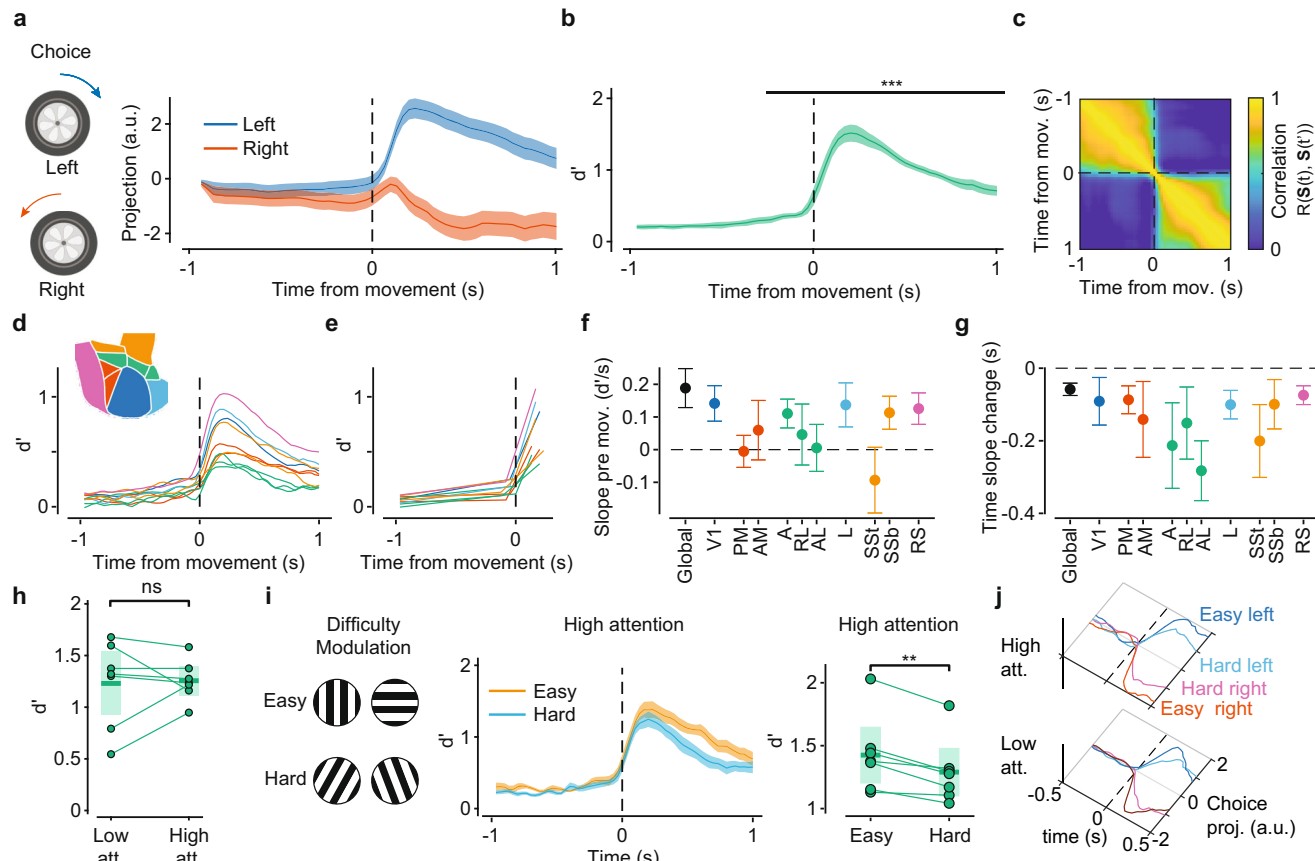

**Fig. 3 | Choice signals have pre-motor component and are modulated by task difficulty and attention. a** Projections of left- and right-choice trials on the choice state axis for a characteristic animal (line for trial average and shaded area for its 95% CI). Wheel movements signaling either a left or a right choice were aligned to the wheel movement onset. **b** Evolution of choice d′ discriminability relative to movement time averaged across animals (n = 7, line for mean and shaded area for its s.e.). d′ values were significantly larger than baseline starting from 0.2 s before movement onset (***p < 0.001 consistently across the range, baseline defined between 1 and 0.5 s before movement onset, two-sided paired t-test). **c** Temporal stability of the state axis for choice, showing a clear change in the contribution of the choice state axis near the time of movement onset. Same animal as in **a**. **d** Temporal evolution of area-specific d′ curves (inset: area color code). **e** Piecewise linear fits of the curves in **d** in pre- and post-movement periods. **f** Pre-movement

slopes fitted in **e** for different areas; error bars, 95% CI of the mean (dots) across animals (n = 7 animals) ("global" indicates multi-area d′). **g** Times of slope change for different areas from the fits in **e**. Data and colors as in **f**. **h** Choice discriminability separated for low and high attention trials. No significant differences were found (p = 0.8, paired two-sided t-test). Dots are different animals; middle lines and shaded areas are means and their 95% CI (n = 7 animals). **i** Left: example stimuli for easy and difficult trials. Middle: evolution of choice discriminability, d′, in high attention states for easy and hard trials (angle difference > or < 45°, line for trial average and shaded area for its 95% CI). Right: Paired comparisons of peak d′ values for each animal (p = 0.003, paired two-sided t-test). Data as in **h**. **j** Temporal evolution of left and right choice projections trajectories (averaged across 48 to 93 trials each), with difficulty modulations and separately for low and high attention trials (representative animal).

detected movement did not always coincide with the movement terminating the trial. However, we confirmed that the direction of the first movement had a large and significant correlation with the trial choice (85 ± 4% agreement with movement directions), suggesting that the decision was made quickly after the stimulus presentation (Supplementary Fig. 7c, d). We then aligned responses relative to movement times and defined a state axis that linearly discriminated clockwise from counterclockwise wheel rotations (hereafter left and right choice, respectively). LocaNMF projections onto this axis sharply separated left from right choices (Fig. 3a), reaching peak separation values approximately 0.15 s after movement detection (Fig. 3b; peak d′ = 1.5 ± 0.1) and with the choice axis showing two clearly stable regions before and after movement onset (Fig. 3c). Area-specific d′ values started to increase from baseline significantly 300 ms before movement onset (p < 0.05, paired t-test, Fig. 3d, Supplementary Fig. 9). We characterized pre-movement components using a piecewise linear regression analysis (Fig. 3e) applied to d′ curves to quantify the slope of the fit before the movement and the time of the slope change (Fig. 3f, g). We found a consistent trend for positive pre-movement slopes (ramping) and pre-movement slope change times (slope = 0.19 ± 0.06 d′/s, p = 0.007,

t-test; time of slope change = −0.06 ± 0.02 s, p = 0.014, t-test), providing evidence for temporally and spatially distributed pre-movement choice components across these regions. The increase in choice discriminability after stimulus presentation was also apparent in trial time, showing a clear split on the trajectories after the open-loop period (Supplementary Fig. 10).

When conditioning on attentional levels, similar discriminability values were found for high- and low-attention trials (Fig. 3h). We reasoned that although evidence accumulation might not be a relevant factor in our task, a decision variable[50]—reflected in the time-varying d′ values—would still retain its sensitivity to task difficulty, more prominently on high attention trials. Indeed, we found that in high-attention states, d′ curves reflected stronger choice separation in easy trials than in difficult trials (Fig. 3i; peak d′ = 1.4 ± 0.1 and 1.3 ± 0.1, respectively; paired t-test, p = 0.003). In low-attention states, a similar trend was observed, but the difference was not significant (paired t-test, p = 0.4). Thus, attention enabled a modulation of the decisional process in proportion to trial-to-trial difficulty changes. An analysis of wheel velocities confirmed d′ modulations did not simply reflect a difficulty-dependent change in motor control (Supplementary

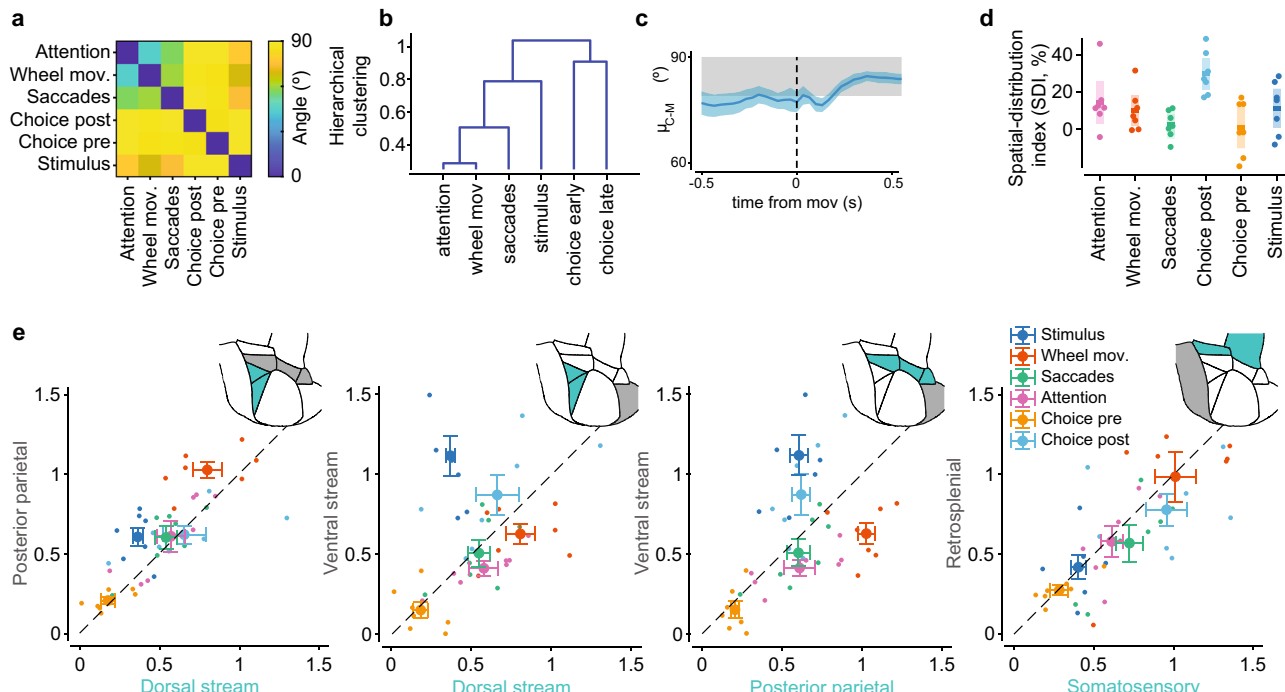

**Fig. 4 | Choice is distributed, near orthogonal to other components and with a ventral-stream dominance. a** Angles between state axes averaged across animals. Choice axes (pre- and post-movement) were orthogonal to all other axes (smallest angle 84 ± 7°). Attention and wheel had the smallest angular separation (44 ± 3°), followed by wheel and saccades (56 ± 4°). **b** Hierarchical clustering from the angle distances in **a**. Attention and wheel movements were most similar. Choice pre- and post-movement onset clustered together, whereas stimulus and saccades had unique profiles. **c** Angle between choice and movement state axes averaged across animals aligned to movement onset (line for average across animals and shaded area for its 95% CI, n = 7 animals); shaded gray band is the expected angular distance range for statistically independent axes; observed angles never significantly

deviated from the statistically independent condition. **d** Spatial-Distribution index (SDI) for each state axis. Choice had the largest SDI (30 ± 4%); dots are different animals; middle lines and shaded areas are means and their 95% CI (n = 7 animals). **e** We computed five d′ values, each derived by restricting locaNMF components to one of the five area groups (insets), thus defining a 5-D space for d′ components. The five broad area groups consisted of the dorsal stream (PM and AM), ventral stream (L), posterior parietal (A, AL, and RL), somatosensory (SSt and SSb) and retrosplenial (RS) regions. Each plot shows a 2-D projection of the five broad area groups, where each dot corresponds to the d′ values for a given animal and the large dot with errorbars to the average across animals and s.e. (n = 7 animals).

Fig. 11a, b). Furthermore, choice axes independently defined in low- and high- attention states were highly correlated (Pearson's r = 0.72 ± 0.03), indicating they reflected a congruent underlying decisional process. We verified that this correlation value reflected a large stability of choice axes across attention states by computing discriminability values in high attention trials using the axis defined during low attention and vice versa (Supplementary Fig. 12). Finally, to examine whether the spatial integration embedded in the locaNMF decomposition was critical for the detection of choice signals, we repeated the d′ analysis, defining state axes based on the activity of individual pixels independently from each other. Both before and after movement onset, this analysis was unable to separate left and right choice trajectories (Supplementary Fig. 11c), in agreement with recent reports relying on a similar independent-pixel analysis[25–27].

Together, these results indicated that choice signals, sparsely distributed in the posterior cortex could be detected through spatial integration; they defined a subspace for left and right trajectories, with a stable representation across states of attention, but with attention enabling difficulty-dependent modulations of response trajectories (Fig. 3j).

**Distinct spatial and temporal characteristics of choice signals**
Distinct signatures of choice signals were evident in the pairwise angular distance between state axes (Fig. 4a). Overall, angles between state axes were greater than 44°, with the choice axis being near-orthogonal to the sensory-, movement-, and attention-related axis. Sensory and movement components had large angles (69 ± 2°, mean + s.e.), and the smallest angles were observed between

the movement and attentional axes (44 ± 4°). In time, choice axes computed separately before (−0.1 s) and after (0.3 s) movement onset were relatively stable in the pre- and post-movement periods and orthogonal to each other (81 ± 3°, with 79°–89° the expected 95% CI for independent axes; Methods; Fig. 3c, Fig. 4a). The angle between wheel movement and saccades axes, similarly, computed across time windows, was also stable with angles of approximately 70° (69 ± 5°; Fig. 4a). Hierarchical clustering analysis on the angular distances (Fig. 4b) highlighted that choice axes pre- and post-movement clustered together and were the most dissimilar to the other state axes.

Irrespective of the time period, choice was nearly orthogonal to the movement axes (Fig. 4c), with no significant differences when comparing with a null model with orthogonal axes, both before and after movement onset (before: 77 ± 3° at t = −0.5 s; p-value = 0.25, one sided t-test against 79° null-model lower bound. After: 80 ± 3° at t = 0.5 s; p-value = 0.73, one sided t-test against 79° null-model lower bound). When transitioning from the pre- to the post-movement period, choice d′ values never collapsed to zero (Fig. 3b), suggesting a rotation of the choice axis while preserving the orthogonality between choice and movement axes. This can be interpreted as a rotating state axis for choice in a multi-dimensional choice sub-space, that remained orthogonal to a similarly defined movement subspace.

We verified that the smaller angles observed between some state axes in Fig. 4a were not a direct consequence of correlated behavioral variables by computing their discriminability power after orthogonalization with the other axes (choice vs attention, movements, and saccades; movements vs saccades and attention; and choice in high

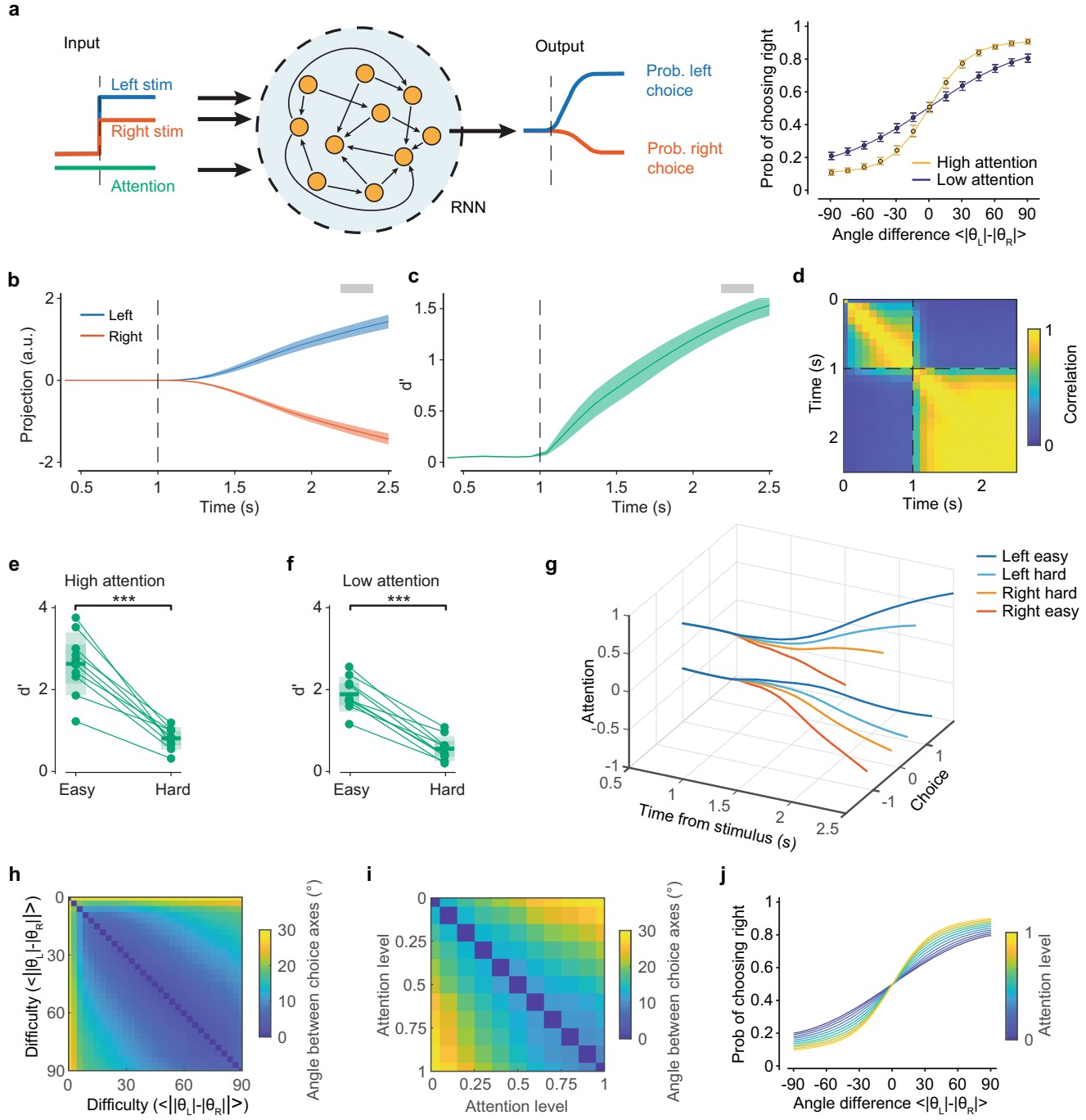

**Fig. 5 | RNN model relates neural representations to DM computations. a** Left: recurrent neural network (RNN) architecture consisting of a module with $N = 50$ recurrently connected units. The module receives two inputs for the left and right stimuli, and one input for the attentional state. It generates a continuous output that will determine the choice. Right: Target psychometric curves—matching the animals' psychometric responses—used to determine the proportion of L/R-choice trials in the training set for each difficulty level, depending on the attention state. Data showing mean and its 95% CI across $n = 200$ trained networks. **b** Projection of left and right trials onto the choice state-axis following the same methods used in Fig. 2. Shaded bar at the top denotes the selected time used for state-axis averaging (line for trial average and shaded area for its 95% CI for a representative network). **c** Evolution of choice axis discriminability over time (line for average across and shaded area for its s.e. across $n = 10$ networks). **d** Cross-correlogram of choice axis

stability averaged across $n = 10$ networks. The choice axis becomes stable quickly after the stimulus presentation ($t = 1$ s). **e** Discriminability (d') in easy and difficult trials during states of heightened attention (group difference significance $p = 1.4 \cdot 10^{-5}$, paired two-sided $t$-test). Dots are different network realizations; middle lines and shaded areas are means and their 95% CI ($n = 10$ networks). **f** As in **e**, but for low attention trials (group difference significance $p = 1.2 \cdot 10^{-6}$, paired two-sided $t$-test). **g** Projected choice trajectories with splits by difficulty and attention. **h** We computed independently choice axes for each difficulty level. Choice axes obtained at different difficulty levels were near parallel to each other, with the largest deviation (between the easiest and most difficult conditions) smaller than 20°. **i** As in h, but for choice axes computed at different attention levels instead.
**j** Psychometric curves from the trained model showing that the model can generalized across levels of attention and difficulty.

and low attention). All state axes retained significant discriminability after orthogonalization (Supplementary Fig. 13).

In addition to these overall representational differences, choice signals also had distinct spatial characteristics relative to other variables. We defined a spatial distribution index (SDI) that captured whether several or only a few areas contributed prominently to the d' discriminability (SDI = 0% if only 1 area contributes and (N-1)*100% if all N areas contribute equally and independently) and found that choice had the largest SDI values (30 ± 4%, more distributed contributions) compared to sensory, movement, and attentional signals (approximately 10%) (Fig. 4d). To further examine the area-specific contributions to choice signals, we clustered higher visual areas into three main groups—ventral (L), dorsal (PM, AM), and parietal (A, RL, AL)[40]—and separately analyzed somatosensory (SSt, SSb) and retrosplenial (RS) regions. V1 contributed an overall uniform d' value to all separations (Supplementary Fig. 3a, b); hence, we did not include it in this analysis of relative differences. We then computed d' values using only the locaNMF components that originated from these grouped areas and did this for all variables: visual, movement, choice, and attention. This resulted in a five-dimensional (5D: ventral, dorsal, parietal, somatosensory, and retrosplenial) space, where the coordinates of a variable reflected the distinct contribution of the grouped areas to the d' separability of that variable. When examining discriminability power in 2-D projections of this 5-D space (Fig. 4e), we could identify area-specific contributions. Stimulus and choice post-movement onset presented the largest contribution in the ventral stream (stimulus, d' = 1.11 ± 0.12; choice post-movement d' = 0.87 ± 0.13). These contributions were significantly larger than those in parietal and dorsal stream areas (ventral vs parietal: $p = 0.012$ and $p = 0.019$ for stimulus and choice; ventral vs dorsal: $p = 0.011$ and $p = 0.043$; paired $t$-test). For the discriminability of attentional states, the parietal and dorsal stream regions had larger contributions than ventral stream regions (high vs low sustained attention, ventral vs parietal: $p = 0.015$; ventral vs dorsal: $p = 0.008$; paired $t$-test). Dorsal and parietal areas contributed the most to the discriminability of movement variables, with the parietal areas having the largest d' for wheel movements (dorsal vs parietal: p = 0.001; ventral vs parietal, p = 0.014; paired $t$-test). Retrosplenial and somatosensory areas contributed similarly to the discriminability of choice and movements, with d' values generally correlated across all variables ($r = 0.79$; 95% confidence intervals [0.64, 0.87], $p$-value $< 10^{-11}$, for d' correlations between somatosensory and retrosplenial areas across animals).

These results highlighted the major differences between choice state axes defined pre- and post-movement. Choice pre-movement was less localized (lower SDI), with each cortical region contributing similarly. On the other hand, choice post-movement was more localized (higher SDI), with larger contributions in ventral stream areas (Fig. 4e). We also computed the increase in choice discriminability from the pre- to the post-movement periods and found that the d' increase from the ventral components ranked significantly higher than the increase associated with components from the dorsal and posterior areas (95% CI) (Supplementary Fig. 14a). Even in the absence of ventral components (i.e., defining the choice axis using only components from non-ventral areas), pre- and post-movement axes were still orthogonal to each other (Supplementary Fig. 14b, Discussion).

In summary, distributed choice signals were distinct from sensory, movement, and attentional components, dominantly in ventral-stream visual areas and modulated by task difficulty and attention, suggesting that they might reflect the decision-making computations associated with the discrimination task.

### RNN modeling of decision dynamics

To examine this possibility, we used RNNs as implementation-level, mechanistic models of the decision-making process. Building on previous work showing that RNNs can capture decision-making computations associated with 2AFC discrimination tasks[51,52], we examined the dynamics of RNNs trained according to the invariance for absolute orientations built into our task—and learned by the animals. Furthermore, rather than using the optimal task solution, we trained RNNs with the trial-to-trial choices of the animals and introduced variability in attention states (Fig. 5a). Using the animals' choices rather than the task rule created numerous contradictory examples, where the input evidence for a left or right choice was non-deterministically associated with left or right output decisions, even in the easiest trials (e.g., non-zero lapse rate). As a result, RNNs learned to produce L/R binary choices from an internal representation that followed a psychometric probability function based on the absolute difference between the two inputs (Fig. 5a). That is, output amplitudes depended only on task difficulty, reflecting a learned invariance for absolute orientations. Context-dependent attention modulations (introduced as an additional binary input) modified output probabilities and created shallower or steeper psychometric curves in low or high attention states, respectively (Fig. 5a right). Performance levels and differences between high and low attention states were chosen to match experimental values. We then analyzed the internal dynamics of the network by computing choice and attention axes from RNNs unit responses, as we did for the neural data with locaNMF components. In the RNNs the choice axis identified a decision variable that represented L/R decisions as separate trajectories in a low-dimensional embedding space (Fig. 5b–d). Furthermore, the separation between L/R trajectories was modulated by attention and task difficulty, with larger separations in easy trials and high attention (Fig. 5e–g). This separation did not depend on absolute orientations, as expected from the RNN having learned this invariance. Attentional modulations maintained an invariant representational geometry of the decision variable across the embedding space, that is, the choice axis remained stable with attention and task difficulty (Fig. 5h, i). This was consistent with what observed in the neural data, where choice and attention axes were near-orthogonal with each other (Fig. 4a). Although the model was trained only with a subset of 13 difficulties and two attention states, it was able to generalize to any difficulty level and range of attention within the trained boundaries (Fig. 5j).

In summary, the similarity of the representations between the RNN and neural dynamics, suggests that the contextually modulated choice signals observed in locaNMF components represented the decision-making computations underlying this task, as learned by the RNN when using the same behavioral output of the animals.

## Discussion

In this study, we used a complex visually guided task to isolate choice signals broadly distributed in the posterior cortex and near-orthogonal to sensory-, movement-, and attention-related variables. We showed that choice signals were prominent in ventral-stream visual areas. Choice signals defined left-right trajectories in a low-dimensional activity space that were modulated by task difficulty, with this modulation enabled contextually by the attention state of the animal. Using RNNs trained on the animals' choices, we showed that the representational dynamics of choice signals were consistent with the decision-making computations underlying the behavioral task. These results suggest a multiplexed representation of variables in the posterior cortex, with a widespread distribution of decisional information, possibly mediating probabilistic inference computations; for instance, information about the ongoing decision-making process could be used for perceptual inference with unreliable sensory stimuli[53,54] and to influence sensory-to-decision signal transformations that inform future action plans. Collectively, our results highlight task and analytical constraints for the detection of choice signals in the mouse posterior cortex, aligning decision-making research in mice during visually guided behaviors with primate studies.

## Methodological relevance

We achieved these results by combining two powerful methods for the analysis of population responses: locaNMF and activity-mode analysis. LocaNMF reduced the dimensionality of the neural data while retaining spatial information, that would have been lost with traditional dimensionality reduction methods (e.g., SVD, NMF). Traditional decomposition methods could also be used to reduce data dimensionality, but the loss of spatial localization would greatly reduce the interpretability power of the components, and the components would also be less robust to session-to-session variability, as previously reported[35]. Furthermore, the state space representation allowed further reduction of dimensionality by aligning the dynamics along task and decision-relevant dimensions. This latter step took place within an interpretable linear framework, where the angle between the state axes as well as d' values could be directly linked to the linear discriminability of the underlying variables.

Dimensionality reduction via trial and spatial averaging permits the robust isolation of state axes, however, it can also collapse subspaces over which important aspects of the dynamics might evolve. For instance, we found that pre- and post-movement decision axes had large angular separations, defining a sub-space orthogonal to that of movement signals. We also found that the dimensionality of choice signals was further enriched by attentional modulations, which produced an isomorphic shift of choice representations. It is conceivable that extra dimensions linked to more subtle aspects of the decision-making process exist but have been collapsed by averaging signals across trials. Future investigations, for example based on probabilistic low-rank dimensionality reduction methods[55], might be able to uncover these hidden dimensions.

The imaging methodology and data analysis used in this study facilitated the identification of distributed choice signals encoded by sparse populations of cells[56]. Indeed, sparse encoding in multi-region networks would make choice signals hard to detect with methods that examine decision information independently at each cortical location, whether because of the use of a single electrode (or multi-contact electrode shank)[24] or because of imaging data analysis focused on individual locations (pixels) independent from each other[26,27,57]. We confirmed this observation by reanalyzing our data at the single pixel level: assuming independence between the activations of different pixels, we failed to detect choice information both before and after movement onset. This result may explain why some recent mouse studies have failed to detect choice signals in posterior cortices during similar visually guided tasks[24–27].

Our choice of seeding-area sizes was chosen to approximately match the spatial correlation length of visual or movement components using smooth penalty boundaries, with sharp boundaries imposing an unnecessary split of components at the border between seeding areas. Small changes in seeding parameters (areas size, centers, smoothness of the boundaries) did not significantly affect the properties of the state axes, suggesting that the spatial correlation length of broadly distributed choice signals approximately matched that of visuomotor components. Additionally, as previously reported[35], the robustness and stability in spatial profiles across experimental sessions enabled by this seeding method follows from two main observations: 1) the boundaries between visual areas are identified by gradient flips in field sign maps, but the identification of the boundaries can be imprecise, especially for the smaller higher visual areas (as we observed in our data and as discussed, e.g., ref. [58]). Allowing a seeding boundary to extend beyond retinotopic boundaries reduces the dependence of the results on the precise identification of these boundaries. 2) Activity components do not have to follow retinotopic boundaries (e.g., if representing afferent signals from other brain regions or top-down modulations), but these activations are still likely to be spatially correlated. A loose (penalty-based) boundary component allows to capture spatially coherent signals crossing area boundaries.

Another contributing factor to the identification of choice signals could have been task complexity. In our task, mice were asked to make a relative comparison between stimulus orientations, a difficult task even for primates[17,59], whereas other studies used simpler visual detection[27] or contrast discrimination tasks[25]. More complex perceptual decisions engage more spatially distributed networks;[60] therefore, the complexity of our task might have facilitated the emergence of choice signals in these posterior cortical regions.

## Feedback origin of choice signal

Choice signals emerged after stimulus onset, were broadly distributed in the posterior cortex, and could be significantly detected as early in the visual hierarchy as in V1, suggesting feedback activations from areas causally involved in the decision-making process. Other non-sensory signals identified in our recordings, (e.g., related to body and eye movements), could also result from feedback activations. Indeed, feedback signals to the posterior cortex have been extensively documented in the literature, in association with a great diversity of underlying variables and computations, including attentional modulations[61], movement-associated responses[21], sensory context[62], and predictive coding[63].

## Choice signals have distinctive spatial and temporal signatures

The properties of the choice signals met several criteria that are characteristic of a decision variable. Their pre-movement components suggested that they did not simply reflect the execution of a motor plan or an unsigned pre-motor preparatory state[64]. Choice signals did not simply reflect bottom-up, stimulus-related information that correlated with the decision process because, given the task design, the contralateral stimulus orientation was uninformative for L/R decisions[17]. Furthermore, at the mesoscale resolution used in this study, we showed orientations were not decodable from the neural signal[65]. In addition, choice signals were modulated by task difficulty, with the strength of the modulation dependent on the attentive state of the animal.

Choice signals could be separated from movements. The cortical localization of movement components was prominent in dorsal-stream regions, consistent with previous reports[20]. Choice signals were instead localized in the retrosplenial cortex and in the visual cortex, mostly in ventral stream regions, along the so-called "what" visual pathway[66]. This was consistent with the task requirements: mice had to evaluate the orientation content of both stimuli and make relative orientation comparisons. Absolute orientations were uninformative, as were the locations of the stimuli, which were unchanged across trials; thus, in contrast to the "where" type of information, which is supposedly associated with dorsal stream regions, solving the task relied on "what" information in ventral stream areas.

We also verified that ventral stream responses were not linked to eye movements (Supplementary Fig. 15a), which typically followed whole-body movements[20], or to stimulus movement (Supplementary Fig. 15b) However, signals detected in ventral areas may still be associated with motor-related components that also carry choice-relevant information[67]. The fact that pre- and post-movement choice axes remained orthogonal to each other in the absence of ventral components (Supplementary Fig. 14b) suggests a representational change of choice information broadly distributed in the posterior cortex and not restricted to ventral regions. What, mechanistically, could cause this change is difficult to establish solely from our widefield data. It could reflect a change in afferent choice signals from the pre- to the post-movement periods: for example, before the movement, choice signals could reflect a distributed afferent from midbrain regions[24], while post-movement, choice-related components from motor cortices[67,68] could add to those from the midbrain and possibly with a stronger signature along ventral regions. Future studies examining functional feedback to these regions in similar decision-making tasks might shed light on these mechanisms.

Attention-mediated modulations were identified via an analysis of pupil area changes around the time of stimulus onset, known to reflect variability in internal states of the animals due to changes in engagement and vigilance during the task, with sustained attention referring to a broad spectrum of these goal-directed internal states[20,47,48,69]. In the time window of our analyses, the main event triggering changes in pupil area was the stimulus onset. The variability in pupil area changes correlated with task performance and response times (Supplementary Fig. 7a, b) and could also be associated with sustained changes in cortical activity, as demonstrated by the presence of a stable attention axis throughout trial time (Supplementary Fig. 3d). The axes' stability was not linked to trial outcome, which would have been the case if the changes in pupil area were related, for example, to reward delivery or other task variables (Supplementary Fig. 8).

Attention-mediated modulations were orthogonal to the subspace defined by choice variables, with the choice axis remaining significantly autocorrelated across time irrespective of attentional state. This can be described as an isomorphic transformation in the embedding space of the decision variable, where the subspace defined by the L/R trajectories is shifted without deforming the representational geometry. The modulation of the decision variable with task difficulty was clear in high-attention trials, but not significant in low-attention states. This might reflect an actual dependence of the decision-making process on attention, given that mice might commit to a difficulty-independent heuristic strategy in low-attention states[17].

The analysis of angles between state axes highlighted a large angular separation between variables, with choice and movements remaining orthogonal to each other before and after movement onset. However, movement onset correlated with a 90° rotation in the choice axis while retaining orthogonality with movements. This phenomenon can be interpreted as movement signals affecting the multi-area dynamics as an endogenous contextual input triggering a rotational dynamic in a multi-dimensional choice space[52,55]. A similar rotational dynamic was observed for movements signals in a movement space, but the subspaces defined by choice and movement rotations remained orthogonal to each other. The large angular difference between pre- and post-movement axes indicated distinct preparatory and movement response signatures in the posterior cortex, possibly reflecting a similar distinction in anterior motor regions[64,70]. Movement signals potently increased neuronal activity across multiple posterior regions[19,20]. These large activations occupy the main dimensions of variability[22], enabling a representation shift while keeping movement representations separate from other variables, that is, in near-orthogonal subspaces. Cellular-level simultaneous recordings from motor areas and posterior cortical regions may provide further evidence for this interpretation in future studies.

Sensory and movement axes had the smallest separation. This latter observation agreed with previous reports both at the local scale of small neuronal assemblies[22] and at the mesoscale[20], indicating a covariability axis between these variables. Similarly, the smaller angles observed between the movement axis and the attentional axis agreed with a recent report showing that attention enhances distinctive spatial features in movement-related activations across these cortical regions[20]. However, even after axes orthogonalization, decoding attentional and movement information was still possible.

## RNN implementation and mechanistic insights

Recurrent state-space models, including RNNs, have been previously used in mechanistic investigations of decision-making processes[51,52]. Moreover, analyses of the similarity of the state-space representations in RNNs and neural responses has been successfully used to infer underlying computations[52]. Here, we adopted a similar approach, but with three main distinguishing features. First, we trained the network with the animals' decisions, rather than the task rule. This constituted a relevant departure from previous research, which added noise to fully deterministic RNNs to capture logistic behavioral tuning functions[51,71]. Instead, we trained the network with contradictory information, such as that involved in the inconsistent trial-to-trial animals' decisions, thus exposing the network to the biases and heuristics of the animals. This allowed the network to capture the probabilistic choice behavior of the animal, agnostically relative to the causes underlying the animals' trial-to-trial variability. Training with the animal choices was akin to training with label noise, for which many deep learning algorithms are robust[72]. The RNN outputs effectively implemented two dynamic accumulators providing time-dependent scores for L/R choices, with the difference between the scores being proportional to the psychometric function. This result was probably related to the mathematical observation that if L/R choices were determined by two accumulators (for the left or right evidence, respectively), the log-likelihood ratio of the conditional probabilities for a given choice, given the state of the accumulators, can be shown to be proportional to the psychometric (logistic) function[73,74]. The temporal dynamics of the RNN enabled a representational comparison with the time-evolving neural trajectories, but it was not intended as a mechanistic descriptor of a decision time. The second novelty was that we trained the RNN to learn an invariance regarding absolute orientations, which were uninformative for the task choice and that was readily learned by the network. Finally, the third novelty concerned attentional modulations. As observed in the neural data, the added attentional input to the RNN caused an isomorphic shift of the decision-making manifold, which retained the geometry of the decision variable. Geometry-preserving isomorphic shifts in low-dimensional embedding spaces, might reflect a general decorrelation principle for variables that are concurrently represented across overlapping cortical networks[75]. These results confirmed that, mechanistically, the representational dynamic of choice signals reflected the decision-making computations underlying the animals' psychophysical behaviors.

## Limitations and open questions

Our results raise several questions to be addressed in future studies; for instance, whether the broad distribution of choice signals mirrored equally broad spiking activations is still unclear. Regarding anatomical considerations, feedback signals are known to preferentially target deep layers (five and six) and layer one[76]. Considering that our imaging macroscope focused on superficial cortical layers and that GCaMP was expressed across the cortex, choice signals might reflect long-range axon-terminal activations and /or depolarizations in apical dendritic trees rather than somatic firing[77]. Concurrent imaging and electrophysiological recordings across layers would clarify this point.

Our study relied on correlative measures; therefore, loss- and gain-of-function perturbative experiments will be necessary to establish causality. Of particular interest would be the inactivation of lateral visual areas in view of the observed ventral-stream prominence. Furthermore, patterned optogenetic methodologies with single-cell resolution might enable the stimulation of the individual neurons that most significantly carry choice-relevant information in these regions to examine their causal contributions to the animals' choices.

Our study focused on features of choice signals that were stable and consistent relative to the temporal structure imposed by our task design. However, it is very likely that other task-uninstructed components (e.g., motor, attentional, decisional) might exists, and more in general, components that do not bear a systematic temporal relation with the trial structure, and therefore characterized by a large trial-to-trial timing variability within and across trials. Our temporal alignment and trial-averaging procedure would average-out these components, thus reducing the effective degrees of freedom of the representations. In summary, broadly distributed decision signals, with a representational dynamic consistent with decision-making computations underlying our visually guided perceptual task, represent a computational substrate capable of modulating early sensory

processing and sensory to decision transformations. These modulations, depend on the underlying decision-making process and might involve probabilistic-inference computations in changeable agent-environment interactions[78].

# Methods

## Experimental procedures
Details of the experimental procedures (surgeries, behavioral training, recordings of body and eye movements, imaging methods, and pre-processing of fluorescence data) have been described in Abdolrahmani and collaborators[20]. We summarize them here in brief.

## Surgeries
All surgical and experimental procedures were approved by the Support Unit for Animal Resources Development of RIKEN CBS. The transgenic mice used in this work were Thy1-GCaMP6f mice ($n = 7$, 6 male 1 female). For all reported results, the number of valid sessions per animal ranged from 11 to 35, with a minimum and maximum number of trials per animal of 2000 to 5000. The Animals were implanted with cranial posts for head fixation and a round chamber consisting of two overlapping glass coverslips (6 mm diameter) for optical access to neural recordings.

## Behavioral training
Animals were trained on a 2AFC orientation discrimination task following the iterative protocol presented in[17]. The final stage of the task consisted of two oriented Gabor patches shown to the left and right side of a screen positioned in front of the animals at ± 35° eccentricity relative to the body's midline. Mice had to report which of the two stimuli matched a target orientation (vertical, $n = 5$; horizontal, $n = 2$). The smallest orientation difference was 9° except for one animal where it was 3°. The largest difference—the easiest discrimination—was ± 90°. Animals signaled their choice by rotating a rubber wheel with their front paws, which shifted stimuli horizontally on the screen. Every trial consisted of an open-loop period (OL: 1.5 s) starting after stimulus onset, during which wheel rotations did not produce any stimulus movement, and a closed-loop period (CL) starting after the end of the OL and lasting 0–10 s, followed by an inter-trial interval (ITI: 3–5 s randomized). For a response to be correct, the target stimulus had to be shifted to the center of the screen, which led to the animal being rewarded with 4 μL of water. Incorrect responses were discouraged with a prolonged (10 s) inter-trial interval and a flickering checkerboard stimulus (2 Hz). If no response was made within 10 s (time-out trials), neither reward nor discouragement was given. Animals were imaged after exceeding a performance threshold of 75% correct rate for 5–10 consecutive sessions. To work with a coherent behavioral dataset, we excluded sessions with exceedingly large fractions for time-outs (≥ 20%) or with average performance below 60%. We recorded cortical responses, wheel rotations, and eye/pupil videos from a 1 s pre-stimulus period until the end of the trial.

## Saccades, pupil area, and body movements
We monitored the contralateral eye using a CMOS camera. Automatic tracking of the pupil position was done with custom software (MATLAB r2020a, Mathworks®). We confirmed the accuracy of pupil tracking by visually observing hundreds of trials. Saccade detection was achieved by applying an adaptive elliptic thresholding algorithm to saccade velocities (as detailed in ref.[20]). We discarded saccades that lasted ≤ 60 ms and were smaller than 1.5°. We extracted the time, magnitude, duration, velocity, start and landing positions of each saccade. We calculated the average pupil area for each imaging session by averaging area values across all trials within the session. Pupil area amplitudes in every trial were z-scored, centering values relative to the session mean.

## Wheel detection
We recorded wheel rotations with a rotary encoder attached to the wheel and flagged as potential wheel movements the time points when the velocity had a zero-crossing (i.e., a sign change) and deviated from zero above a fixed threshold (20°).

## Imaging
Mice were placed under a dual cage THT macroscope (Brainvision Inc.) for wide-field imaging in tandem-lens epifluorescence configuration using two AF NIKKOR 50 mm f/1.4D lenses. We imaged GCaMP6f fluorescent signals using continuous illumination and a CMOS camera (PCO Edge 5.5) with acquisition speeds of either 30 or 50 fps. Illumination consisted of a 465 nm centered LED (LEX-2, Brainvision Inc.), a 475 nm bandpass filter (Edmund Optics BP 475 × 25 nm OD4 ø = 50 mm) and two dichroic mirrors with 506 nm and 458 nm cutoff frequencies, respectively (Semrock FF506-Di03 50 × 70 mm, FF458-DFi02 50 × 70 mm). Fluorescence light path travelled through the two dichroic mirrors (458 and 506 nm respectively) and a 525 nm bandpass filter (Edmund Optics, BP 525 × 25 nm OD4 ø = 50 mm).

## Pre-processing of fluorescence data
GCaMP data was registered automatically using Fourier-based subpixel registration[79]. To compute relative fluorescence responses, we calculated a grand-average scalar $F_0{}^{i,j} = <I_{x,y,t}^{i,j}>_{x,y,t}$, with $I_{x,y,t}^{i,j}$ representing the XYT image tensor in trial $i$, session $j$. We then used this scalar to normalize the raw data tensor $F_{x,y,t}^{i,j} = (I_{x,y,t}^{i,j} - F_0{}^{i,j})/F_0{}^{i,j}$. The data for each trial were then band-pass filtered (0.1–8 Hz). Each tensor was compressed with spatial binning (130 × 130 μm² with 50% overlap). Trial data recorded at 50 fps was further downsampled to 30 fps.

## Hemodynamic correction of fluorescence data
In recordings to control for the hemodynamic signal, we followed a previously reported methodology[44,80]. Briefly, cortical tissue was illuminated at 60 Hz with 15 ms exposure by interleaving shutter-controlled blue and violet LEDs. Blue light path consisted of a 465 nm centered LED (LEX-2, Brainvision Inc.), a 475 nm bandpass filter (Edmund Optics BP 475 × 25 nm OD4 ø = 50 mm) and two dichroic mirrors with 506 and 458 nm cutoff frequencies, respectively (Semrock FF506-Di03 50 × 70 mm, FF458-DFi02 50 × 70 mm). The violet path consisted of a 405 nm centered LED (Thorlabs M405L2 and LEDD1B driver), a 425 nm bandpass filter (Edmund Options BP 425 × 25 mm OD4 ø = 25 mm), a collimator (Thorlabs COP5-A) and joined the blue LED path at the second dichroic mirror. Fluorescence light path travelled through the two dichroic mirrors (458 and 506 nm respectively) and a 525 nm bandpass filter (Edmund Optics, BP 525 × 25 nm OD4 ø = 50 mm) and was captured with a PCO Edge 5.5 CMOS camera with cameralink interface. Camera acquisition was synchronized to the LED illumination via a custom Arduino-controlled software. Frame exposure lasted 12 ms starting 2 ms after opening each LED shutter.

Continuously acquired imaging data was then split into blue and violet channels and registered independently to account for motion artifacts. For every pixel blue and violet data was independently transformed to a relative fluorescence signal, $\frac{\Delta F}{F} = (F - aF - b)/b$, where $F$ is the original data and $a$ and $b$ coefficients are obtained by linear fitting each timeseries, i.e., $F(t) \sim at - b$. Afterwards, for each pixel, violet $\frac{\Delta F}{F}$ signal was low-pass filtered (6th order IIR filter with cutoff at 5 Hz) and linearly fitted to the blue $\frac{\Delta F}{F}$ signal: the hemodynamic-corrected $\frac{\Delta F}{F}$ signal was obtained as $\frac{\Delta F}{F}$ corr $= \frac{\Delta F}{F}$ blue $- (c\frac{\Delta F}{F}$ violet $+ d)$, where $c$ and $d$ are the coefficients from linearly fitting the low-pass filtered $\frac{\Delta F}{F}$ violet to the $\frac{\Delta F}{F}$ blue signal, i.e., $\frac{\Delta F}{F}$ blue$(t) \sim c\frac{\Delta F}{F}$ violet$(t) - d$. Finally, $\frac{\Delta F}{F}$ corr was low-pass filtered (6th order IIR filter with cutoff at 8 Hz) and spatially downsampled such that every pixel measured 50 × 50 μm².

## Retinotopies

We used a standard frequency-based method (Kalatsky and Stryker, Neuron 2003) with slowly moving horizontal and vertical flickering bars and corrections for spherical projections[81]. Visual area segmentation was performed based on azimuth and elevation gradient inversions. Retinotopic maps were derived under light anesthesia (Isoflurane) with the animal midline pointing to the right edge of the monitor (IIYAMA Prolite LE4041UHS 40") and the animal's left eye at a distance of 35 cm from the center of the screen.

## Alignment to the Allen mouse brain common coordinate framework

Imaging data from each animal was aligned to the Allen Mouse Brain Common Coordinate Framework (CCF) following the approach described by Waters[82]. In brief, we extracted the centroids of areas V1, RL and PM, using them to create a triangle that we aligned to the one from the Allen CCF. We did so by first making the V1 vertices coincide and then rotated and scaled the triangle to minimize the distance between the other vertices while maintaining the original angles.

## Data processing

**LocaNMF.** LocaNMF analysis was conducted following the methods described by Saxena et al.[35]. Imaging data across all trials and sessions was first concatenated and its dimensionality reduced using singular-value decomposition (SVD) up to 99% of the original variance. LocaNMF was initialized using 10 regions based on the Allen CCF and centered on V1 (VISp), PM (VISpm), AM (VISam), A (VISa), SSt (SSp-tr), RL (VISrl), SSb (SSp-bfd), AL (VISal), L (VISl and VISli), and RS (RSPagl and RSPd) (Supplementary Fig. 1), with regions extending beyond retinotopically-defined area boundaries. For each region, a spatial mask was created by setting a distance $D = 1$ within the region boundaries and an exponential decrease (to zero) for pixels outside the boundary. The localization penalty for each pixel was 1-D (Supplementary Fig. 1a). LocaNMF rank line search was run for these 10 regions with a localization threshold of 75% and total explained variance of 99%, resulting on average in approximately 200 components per animal (an example decomposition is shown in Supplementary Fig. 2). After decomposition, temporal components were split back into the original trial structure. More formally, LocaNMF produced a decomposition tensor $F^i_{x,y,t} \sim \sum A_{x,y,k} C^i_{k,t}$ for trial $i$, where $A_{x,y,k}$ is the spatial part of component $k$, and $C^i_{k,t}$ is its temporal part. Since locaNMF is based on NMF, components originating from a given region are not orthogonal to each other. However, components originating in different regions had minimal overlap due to the initialization process mentioned above. The spatial components of the decomposition were significantly localized and could be mapped onto the original seeding region. The temporal component captured the unique trial-to-trial variability, and all subsequent analyses in the time domain were conducted using only the temporal $C^i_{k,t}$ of locaNMF components. Analysis of components' role were always performed on a per-area basis (i.e., using all components of a given area), rather than on single components. The total explained variance (EV) of a partial decomposition was computed, for each pixel, by computing the variance of $\sigma^2(\sum_{k \in S} A_{x,y,k} C_{k,t})_t$, where S is a subset of components, and where all trials have been concatenated to produce $C_{k,t}$. The total variance is then summed across all pixels and compared with the variance of the original $F_{x,y,t}$ signal.

## State axis definition

We defined a global state axis as a one-dimensional projection of locaNMF temporal components $C(t)$ that maximized the weighted distance between the trajectories of two trial groups **A** and **B** (bold letters indicate vectors). For each group, we defined trial-averaged trajectories $\langle A(t) \rangle$ and $\langle B(t) \rangle$ and defined $S(t) = \| \frac{\langle A(t) \rangle - \langle B(t) \rangle}{\sigma_{AB}(t)} \|$, where

$\sigma_{AB}(t) = \sqrt{\frac{1}{2}(\sigma^2_{A(t)} + \sigma^2_{B(t)})}$ is the pooled standard deviation between the two groups. State axis projections for the i-th trial were then obtained by the dot product $P^i(t) = S(t) \cdot C^i(t)$, where $C^i(t)$ are the trial-averaged temporal locaNMF components of trial $i$. Because there is no mean-centering in time of the components, i.e., $\langle C^i(t_j) \rangle_i \neq 0$, activity changes across all areas can cause co-fluctuations in the projected trajectories. Mean-centering of projections was performed on 3D representations for visualization purposes (Fig. 3j, Fig. 5g). Discriminability between the original A and B groups was then computed as $d' = \frac{\langle P^i_A(t) \rangle - \langle P^i_B(t) \rangle}{\sigma_{P_A P_B}(t)}$, that is the difference between the averaged projections of groups A and B, divided by their pooled standard deviation. To validate state axes projections and discriminability we performed five-fold cross validation: state axes were defined using only 20% of the trials of each group and projections and d' were computed on the remaining trials. To compute and validate the state axes, both groups **A** and **B** always had the same number of trials (i.e., the number of trials in the smallest group and a random set of the same size from the other group). The number of trials used to define all the state axis used in the manuscript are shown in Supplementary Table 2.

## Area-specific state axes

We defined state axes for each of the 10 areas by only using the subset of locaNMF components $C^i_{k,t}$ that originated from that area. This was akin to first projecting onto the subspace defined by the components of a particular area, and then obtaining the associated state axis.

## State axes stability

To assess the stability of state axes, we used our original definition of the time-dependent state axes, that is, using components from all the areas, and a "backward" three-frame averaging window (around 100 ms) and then computed its temporal autocorrelation $C(S(t), S(t'))$. For sensory, movement, and sustained attention state axes we chose the time-independent state axes $S \equiv S(t^*)$, where $t^*$ was chosen from the largest stability cluster (represented by a gray bar in the respective figures). For the state axis of choice, we used the original $S(t)$ to monitor when choice information first appeared and whether its signature was unique.

## Task-related state axes

**Stimulus.** For stimulus state-vectors, we used for the first group (**A**) trials in the time interval (−0.5 to +0.5 s) centered on the stimulus onset. For group **B**, since the stimulus was present in all trials, we used the same trials in the preceding time interval (−1 to 0 s) as the no-stimulus condition. Since the preceding ITI was randomized (3-5 s), time-dependent contributions to the signals should be similar across the two groups.

**Contralateral stimulus.** We used trials with the left stimulus horizontal as group **A** and the trials with the left stimulus vertical as group **B** (Supplementary Fig. 4).

**Wheel movement.** Group **A** consisted of trials for which the first movement after stimulus presentation occurred at least 0.5 s after stimulus onset and without any saccade detected in the previous 0.5 s. These trials were aligned to the detected movement onset. Group **B** consisted of trials with no movement detected during the first 5 s after stimulus onset. These trials were aligned with respect to a frame picked at random within the same time interval.

**Saccades.** Akin to the definition of wheel movements, Group **A** consisted of trials for which a saccade after stimulus presentation occurred at least 0.5 s after stimulus onset and without any wheel movement

detected in the previous 0.5 s. These trials were aligned to the detected saccade time. Group **B** consisted of trials with no saccade detected during the first 5 s after stimulus onset. These trials were aligned with respect to a frame picked at random within the same time interval.

**Sustained attention.** Sustained attention was measured by changes in pupil area during the stimulus presentation. We computed pupil area changes (pA) as the difference between the maximum pupil area during the open-loop period (i.e., the 1.5 s window after stimulus onset) and the average area 1 s before stimulus onset. We labeled as "high sustained attention" trials (group **A**) those in the top 33rd percentile of the pA distribution and as "low sustained attention" trials (group **B**) those in the bottom 33rd percentile. Groups **A** and **B** were always balanced by definition.

**Choice.** L/R choices in each trial were measured from the direction of the first movement after stimulus onset. The state axis for group **A** was computed from right-choice trials and for group **B** using left-choice trials. As for the detection of wheel movements, we restricted the analysis to trials in which the first movement occurred at least 0.5 s after stimulus onset and with no saccades 0.5 s before the detected movement. Trials were aligned to the time of movement detection.

**State axes independence**
To determine confidence intervals of the angle formed between independent state axes (Fig. 4c), we proceeded as follows. For each animal, we picked at random one of the original state axis, and sampled with repetitions from its components to generate 2000 surrogate state axis with matched dimensionality. The angle between the original vector and each of the surrogates was computed, resulting in an angle ranging from 79° to 89° (2.5% and 97.5% confidence intervals).

**Piecewise linear fitting of d' curves**
To fit the time-evolving d' curves to periods before and after movement onset, we performed two-slope piecewise linear fitting using the Shape Language Modeling toolbox (MATLAB Central File Exchange, John D'Errico, 2021; SLM−shape language modeling; https://www.mathworks.com/matlabcentral/fileexchange/24443-slm-shape-language-modeling). This method performs two linear fits in a fixed interval with a single knot between them. We chose the interval −1 s before movement onset up to the 95th percentile of the peak post-movement response amplitude (typically occurring around 0.3 s after movement onset). The position of the knot determined the slope change time.

**Spatial-Distribution Index (SDI).** The SDI for a given state axis was computed as $\mathrm{SDI}(\%) = \left(\frac{d'_{\mathrm{global}}}{\max(d'_i)} - 1\right) \cdot 100$ where $d'_i$ refers to the area-specific state axes d' scores (i.e., a state axes defined using only the weights of i-th area), and $d'_{\mathrm{global}}$ refers to the discriminability of the original state axis (i.e., the state axis defined using all components). Since the original state axis uses all components across the areas, $d'_{\mathrm{global}}$ is an upper bound of $d'_i$. Similarly, each $d'_i$ contributes sublinearly to $d'_{\mathrm{global}}$, since correlated activity across areas would not result in higher discriminiability. Hence, SDI measures how much the discriminability can increase when using all areas vs just the one with the largest d'. The SDI plays a similar role to deviance-explained in statistical models, as it measures the relative contribution of a small model (single area) vs the full model (all areas). In this definition, SDI = 0% if only 1 area contributes and (N-1)*100% if all N areas contribute equally and independently.

**Pixel-wise choice decoding.** To compute choice-related d' values for individual pixels, we proceeded as follows. Using the same groups of trials described in "Task-related state axes−choice", at each pixel and time (relative to movement onset) we computed the mean and variance of the dF/F distribution for each group. We then computed d' with the

usual formula as $d'^{i,j}(t) = \frac{<dF/F_A^{i,j}(t)> - <dF/F_B^{i,j}(t)>}{\sigma dF/F_A dF/F(t)}$, where $i,j$ denotes the i-th row and j-th column of the original dataset. This calculation was restricted to the pixels that were common across all animals (pixels at the edges of the imaging window were not present in all animals due to different alignment transformations to the reference dataset).

**RNN model**
The RNN consisted of a single RNN module with $N = 50$ neurons (ReLU activations), receiving 3 inputs (left stimulus, right stimulus, and attention level) and producing a binary response as an output for left or right choices (softmax activation).

**Inputs.** The input space consisted of a sequence of 25 frames. Stimulus orientations were mapped to the range of −1 to +1 (corresponding to −90° to +90°) and were presented after the first 10 frames. The difficulty of a trial was encoded by the absolute difference between the two stimulus signals. Attention was modeled as a constant binary signal (0 or 1), already present at the beginning of the trial. A small noise (normally distributed with amplitude 0.1) was added to the input signals to improve the robustness of the optimization, but it was irrelevant for the psychometric fitting.

**Training.** For training the network, we generated simulated animal responses by computing L/R choices, following a psychometric curve of the form $P_{\mathrm{left}}(\theta) = \frac{1}{1+e^{-\alpha\theta}}(1-\lambda) + \frac{\lambda}{2}$, where $\theta$ is the difference between the two inputs, $\lambda$ is the lapse rate, and $\alpha$ controls the slope. We used a constant $\lambda = 0.2$ and $\alpha = 2/90$ for low attention and $\alpha = 5/90$ for high attention. Lapse rate was chosen to match that of a representative animal. Slopes were chosen to match average performance across difficulties of a representative animal (67% and 80% for low and high attention respectively, sampled from a uniform trial difficulty distribution). To train the network, we used 6400 trials per difficulty level and chose 13 difficulty levels with angle differences uniformly distributed from −90° to +90°. We trained the network using a batch normalization layer and a custom loss function consisting of the categorical cross entropy at the time of stimulus presentation and at the last frame. The output was a binary vector with three components (L,R,N), representing left, right, and no choice conditions. At the time of stimulus presentation, the vector was set to (0,0,1) and at the trial end to (1,0,0) or (0,1,0). We included the stimulus presentation time and no-choice condition in the loss function to prevent the output drifting before stimulus presentation, following a procedure used by Mante (Mante et al., Nature 2013). Accuracy during training was computed using the categorical accuracy at the end of the trial. The network was implemented with TensorFlow 2.0 and trained using the Adam optimizer for 25 epochs with a batch size of 640. Note that training the network with the animal choices made the network robust to overfitting. We trained 10 different networks (200 for Fig. 5a) by generating new sets of inputs and randomly initializing the network weights.

**Analysis.** We analyzed the output of the RNN in the same way as for the neural data, but we used the time series of the $N = 50$ neurons instead of the locaNMF components to define choice and attention state axes.

**Reporting summary**
Further information on research design is available in the Nature Portfolio Reporting Summary linked to this article.

# Data availability
Source data are provided with this paper. Data required to replicate the findings is available through ZENODO (CBS-NCB/distributedDM: Public release (v1.1). Zenodo. https://doi.org/10.5281/zenodo.7435887). Unprocessed raw data is too large for permanent external

storage and will be made available by the corresponding author upon reasonable request. Atlas reference data for brain areas alignment across animal is obtained from The Allen Mouse Common Coordinate Framework[39] (CCF) (https://atlas.brain-map.org/). Source data are provided with this paper.

## Code availability

Code for data processing, analysis and figures is available through ZENODO (CBS-NCB/distributedDM: Public release (v1.1). Zenodo. https://doi.org/10.5281/zenodo.7435887). This code was written in MATLAB (r2020a) and Python (v3.7) with Tensorflow 2.0. Imaging data was collected using custom software based on PCO SDK (v 1.14), eye-tracking data was collected using custom software based on Fly-Capture2 SDK (v8.0), with custom code written in MATLAB (r2020a) to interface with both acquisition systems using MATLAB Data Acquisition Toolbox for National Instruments (NI) devices. Acquisition software is available upon reasonable request.

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

## Acknowledgements

We thank Rie Nishiyama, Yuka Iwamoto, and Yuki Goya for providing technical support for multiple aspects of the experiments. We also thank O'Hara & Co., Ltd. for their support with the equipment. This work was funded by RIKEN BSI and RIKEN CBS institutional funding; HFSP post-doctoral fellowship LT000582/2019 awarded to J.O.; JSPS grants 26290011, 17H06037, 372 C0219129 awarded to A.B.; and a Fujitsu collaborative grant.

## Author contributions

A.B. and J.O. conceived the project and wrote the manuscript; J.O., M.A., R.A., and D. L. collected the data; M.A. and J.O. pre-processed the data and J.O. analyzed the data.

## Competing interests

The authors declare no competing interests.
