## [Peer Review File · Nature Communications]

Distributed context-dependent choice information in mouse posterior cortexREVIEWER COMMENTS

Reviewer #1 (Remarks to the Author):

Orlandi et al study the representation of task-related variables across a network of posterior cortical areas in mice performing an interesting perceptual discrimination task. Activity is measured with mesoscale calcium imaging and analyzed at the population level with dedicated analysis techniques. The primary goals are to identify choice-related activity across posterior cortex, to relate this activity to representations of other task variables (stimulus, attention, movement, saccades), and to reveal differences across areas within posterior cortex.

This goal is important and timely, in particular in light of sometimes contradictory results from past studies. However, the text in the methods/results is at places too compressed to unambiguously understand the details of the employed analysis methods, or to fully grasp the significance of some findings. For this reason, I found it difficult to conclusively evaluate some of the key findings, as I lay out in more detail below.

Main comments:

(1) Local spatio-temporal modes

(1a) How does the spatial distribution of locaMNF components within an area vary across order k ? Are the components orthogonal? If not, retaining only the temporal components $C(t)$ as in line 512 could be problematic. This information should be provided in the methods.

(1b) The authors argue for the use of locaMNF by comparing the resulting classification of task-variables to a method working at the level of individual pixels (Supp. Fig. 8). But why is this the meaningful comparison? Would it not make more sense to compare to the performance of a classifier that operates on the entire state-space of pixels (as opposed to individual pixels)? Such a classifier would have access to images of brain activity, like locaMNF, but would not impose any top-down spatial parcellation along area boundaries.

(2) State axes

(2a) I found it difficult to evaluate some of the results without a better understanding of the structure of the behavior. How correlated are the various behavioral states that are being used to define state-axes? Correlations between behavioral variables may affect the interpretation of the inferred state axes. In particular, do the small angles between state axes in Fig. 4a reflect a property of the neural representations, or do they reflect strongly correlated behavioral variables?

(2b) State axes are defined by comparing two sets of trials/responses. However, in the definition of the stimulus axis, and potentially of other axes as well, the within-trial timing of the two responses is not matched. For example, in Fig. 2d time points in the no stimulus response occurs earlier in the trial than “time-matched” points in the stimulus response. The corresponding stimulus axis could thus potentially reflect “condition-independent” (i.e. time-dependent) variance (see dPCA) in addition to stimulus-dependent variance. Such variance could affect the angle between state-axes.

(2c) Line 525: Stability of state axes. Are the described computations performed separately by area, and stability plots are then averaged across areas? If so, would it not make sense to assess stability at the “whole-brain” level?

(3) Attention vs choice

Fig. 3j provides a summary of the interaction of choice, task-difficulty, and attention. However, were all aspects of that panel verified in the data? For one, task-difficulty is shown as affecting responses along the choice axis, but orthogonal to the attention axis, and the choice axis is shown as being identical in the two attention states. But choice axes estimated separately for low and high attention were not identical (line 178), potentially indicating a component of choice that is not maintained across attention states. Also, d_{prime} for choice seems larger in the low-attention state (Fig. 3h) compared to high-attention, which is not reflected in panel j? The authors should clarify how the summary panel relates to these findings.

(4) RNN modeling

(4a) It is not easy to compare the RNN response to the neural responses, because RNN responses are illustrated with somewhat different figures than the neural responses. For example, in the neural data d_{prime} for choice appears to be larger for low vs high attention (fig. 3h,i). That does not appear to be the case in the RNN (Fig. 5g), although a direct comparison is difficult because the plots are not the same. The authors should simplify the comparison, and clarify what aspects of the neural responses are captured by the RNN, and which are not.

(4b) Line 144: the authors emphasize that the RNN generates probabilistic choices even though it was trained on binary choices. But my understanding of the methods was that the output neurons implement a (probabilistic) softmax output function? If so, is it not a given that the choices of the RNN will describe a smooth psychometric curve?

Minor point

The introduction cites papers across a range of species (primates + rodents), areas (from early sensory to late association and movement), tasks, and methods. While I appreciate the effort at being scholarly and the attempt at linking the presented research to the whole field, some of the references are very brief and compressed. Overall, I found it hard to follow what the authors meant to say, i.e. why a particular reference is used. In essence the authors are providing a review in a few paragraphs, which has the disadvantage of somewhat hiding/obscuring the goals of this study.

Reviewer #2 (Remarks to the Author):

Several studies have shown that the representations of choice information can be found in the occipital, parietal, and frontal cortices, but determining whether these representations across areas are comparable, and separating these choice signals from stimulus and motor information, has been challenging. To investigate choice representation, mice in this study are trained to perform a challenging 2AFC orientation discrimination task during mesoscale wide-field imaging of the posterior cortex. The study employs a tensor decomposition method, locaNMF, to identify unique spatiotemporal components in the imaging data. They then apply an additional dimensionality reduction step, revealing the underlying representation of stimulus, movement and choice information in each area. With support from a recurrent-neural network trained on the animals' behavioral data, this study demonstrates that the posterior cortex contains a multi-area decision variable that is modulated by task and behavioral variables. This exciting study provides new insight into the representation of choice across cortical areas and provides an experimental and analytical framework for further investigations. Overall, the study is clearly presented, and the authors are commended for their multiple experimental and analytical controls throughout. I only have a few suggestions for additional improvement to the transparency of an already excellent study.

1. Figure 1d. The authors should consider using 'Probability right choice' to match with convention, and 'Ignore (%)' instead of 'Time out (%)' since the cause of time out isn't immediately clear.
2. Figure 1e. The authors describe alignment of the multiple higher visual areas with the Allen Institute Atlas using the retinotopic field-sign maps. The authors should provide an example field sign map to show the level of confidence in area assignment.
3. Supplementary Figure 1a. The authors provide penalty maps that describe seeding areas and area boundaries for locaNMF decomposition. What benefit is gained from having boundaries that extend significantly beyond areas? How much does the seeding impact the resulting special components? The authors should address this with additional commentary.
4. Supplementary Figure 1c – On lines 95-97 the authors describe that each main component of a seeding area contributes, on average, 9.6 % of the total explained variance. However, this figure shows that the first component contributes ~45% explained variance. Why is this initial component not addressed? In addition, what conclusions should the reader reach by the comparison of the SVD and locaNMF methods? The figure shows SVD achieving larger explained variance (~80%) than locaNMF with fewer components. Yet, the implication is that locaNMF is a better approach. The authors should unpack this comparison of SVD and locaNMF so the reader can better understand the determining factors in the decision to utilize locaNMF.

5. Figure 2d-g. To provide the readers with an intuition regarding statistical power, the authors should provide details regarding the trial numbers for each particular state axis (i.e. high attention vs low attention).
6. Figure 2g. This figure describes sustained attention based on pupil area changes. How are reward and other known factors that influence pupil size accounted for in this description? The authors should provide additional discussion addressing the illustrated dynamic and the impact of other factors that influence pupil size.
7. Figure 3a. The projection of the right choice is similar to that of the left choice briefly after movement onset, before a reversal. Does this reflect an intrinsic choice bias, or an inability to completely collapse along this state axis? The authors should describe this observation and how dynamics emerge.
8. Figure 3f and 4d. It is not clear how global d' is calculated and what is its relationship to the local d' in each area (it seems like it is sublinear). Similarly, I am confused by the SDI index. It seems like the assumption is that if the max area d' is high relative to the global, then this suggests that it can account for most of the total d' , and therefore it is not evenly distributed. However, this makes assumptions about the linearity of d' summing across areas to create the global d' .
9. Figure 4b – The authors use hierarchical clustering to illustrate the relationship between state axes. The authors note that early and late choice are orthogonal from each other, but make stronger statements about their orthogonality from the other variables. Notably, early and late choice are more different from each other than any of the other variables are different from each other. This result could use some more explanation: what are the major differences between choice early and choice late?
10. Figure 5f & g- There appears to be a discrepancy between these two figures regarding d' amplitudes and timing. Panel g separates the hard and easy trials into high attention and low attention, but after this separation the max d' is lower (~ 5 compared to ~ 7). In addition, the timing of the change in d' in panel g is shifted to the left as compared to panel f. The authors should describe the source of this discrepancy and any implications that it may have on the interpretation of these results.
11. Supplementary figures 5 and 7 have typos in the captions.
 - a. Supp 5 - The caption for panel “c” is labeled “d”.
 - b. Supp 7 – The caption for panel “b” is “As in H, but after movement onset” when it should be “As in a, but after movement onset”.

Reviewer #3 (Remarks to the Author):

The authors recorded wide-field calcium imaging data in mice during a perceptual discrimination task. The task consisted of turning a wheel to indicate which of 2 oriented gratings were most similar to a target orientation. The authors used localNMF to decompose data into spatially-dominated components and then conducted a decoding analysis to establish encoding axes for a variety of task variables.

The experiments were impressively successful and I appreciate the willingness to use mice as a model animal for studying decision making. The analysis was thorough and while most of the authors' claims were substantiated by the data, the authors made some questionable assertions and confusing choices regarding the presentation of the results that I believe can be addressed in revision. The following are specific points that should be addressed.

- The authors used localNMF to decompose the wide-field 2P imaging in to pixels associated with particular regions. I wonder, however, what specifically is gain by forcing the weights to correspond to specific regions. Why not use the global encoding axes and ask post-hoc whether the encoding regions correspond to anatomically-defined regions?
- Figure 3c depicts the cross-correlation between choice axes estimated from different time windows.

As the authors pointed out, the 2 distinct regions suggest that there are 2 stable choice axes. They schematically depict this in Figure 4c. However, they could have simply plotted the projection onto each of these axes and demonstrated the plausibility of their hypothesis, rather than presenting their schematic. A similar visualization could have been done for the movement axes. I would have found this demonstration to be far more convincing.

- Regarding statements starting on line 201, Figure 4c, The authors claim that the choice and movement axes are always orthogonal to each other but it does not appear to be the case based on the inset. In particular, it appears that prior to movement onset there is some non-zero correlation between axes. Could the authors please explain this?

- Figure 3j is a schematic illustration based on the authors' description of encoding geometry in the data and Figure 5h presents this same phenomenon for the data obtained from the trained RNN. It's unclear to me why the authors (again) did not choose to plot the actual data in this way, which seems to me to be entirely possible considering they did so for the RNN, rather than depicting a schematic representation.

- Line 178 "Choice axes independently defined in low- and high-attention states were highly correlated (Pearson's $r = 0.72 \pm 0.03$), indicating they reflected a congruent underlying decisional process." However, $r = 0.72$, while high by some standards, is hardly the same. Can the authors comment on differences in the choice axes between states and explain why these are statistically significantly different?

- Starting on line 260: representational similarity... I am unconvinced that these assertions follow from the evidence provided. First, what properties of the representations found in the RNN should convince us that the mechanism employed by the RNN is the same as that employed by the mice? Why does the lack of neural data lend credibility to the RNN analysis? I am personally left more skeptical by this feature than less.

- The wheel movement axis (Supplementary Fig. 2b) does seem to show a dramatic shift at the time of movement onset but the cross-correlograms suggest a constantly, albeit slowly, changing axis in both the pre- and post- movement periods. This undermines the authors claim that there are 2 movement axes that shift around the time of movement onset. Can the authors please explain?

- Line 344. The authors' comment about "forcing" a shift on other task variables doesn't make sense to me. An encoding axis can shift and remain orthogonal to the other task variable axes without any change in the other axes. This is particularly easy in high dimensions.

- The spatial averaging imposed by localNMF decreases the effective degrees of freedom of the data. Averaging across trials to estimate state axes again reduces the effective degrees of freedom. Can the authors comment on how the effective degrees of freedom of the data could impact the spatio-temporal analysis that they have depicted?

- Line 355: "representational similarity analysis..." The authors here reference Mante et al. (2013) (reference 81). However, I can find no reference to representational similarity analysis (RSA) in this paper. I should point out that RSA is often referred to as a formal analysis method. Perhaps there is some colloquial sense in which the authors are using this term? If so, I suggest the authors choose a different description of this family of analyses.

Reviewer #1

Orlandi et al study the representation of task-related variables across a network of posterior cortical areas in mice performing an interesting perceptual discrimination task. Activity is measured with mesoscale calcium imaging and analyzed at the population level with dedicated analysis techniques. The primary goals are to identify choice-related activity across posterior cortex, to relate this activity to representations of other task variables (stimulus, attention, movement, saccades), and to reveal differences across areas within posterior cortex.

This goal is important and timely, in particular in light of sometimes contradictory results from past studies. However, the text in the methods/results is at places too compressed to unambiguously understand the details of the employed analysis methods, or to fully grasp the significance of some findings. For this reason, I found it difficult to conclusively evaluate some of the key findings, as I lay out in more detail below.

Main comments:

(1) Local spatio-temporal modes

(1a) How does the spatial distribution of locaMNF components within an area vary across order k ? Are the components orthogonal? If not, retaining only the temporal components $C(t)$ as in line 512 could be problematic. This information should be provided in the methods.

We have verified that the spatial distribution of the largest components (by variance explained) remains unchanged within an area when increasing the order k . However, changes are observed on the smaller components for large k s. Since locaMNF decomposition is based on non-negative matrix factorization, the decomposition is not unique. Within each seeding area the components are not orthogonal, as it is true in NMF decompositions, however, the constraint of sparsity between seeding areas ensures that there is minimal shared information between components across areas. We agree with the referee that focusing only the temporal components $C(t)$ could pose an issue if we were to investigate the role of one component at a time. Hence, we always focused on the decomposition and discriminability in a per-area basis (i.e., using all components of a given area together) and not on individual components. These considerations are now explained in the Methods, Line 548:

“Since locaMNF is based on NMF, components originating from a given region are not orthogonal to each other. However, components originating in different regions had minimal overlap due to the initialization process mentioned above. The spatial components of the decomposition were significantly localized and could be mapped onto the original seeding region. The temporal component captured the unique trial-to-trial variability, and all subsequent analyses in the time domain were conducted using only the temporal $C_{k,t}^i$ of locaMNF components. Analysis of components’ role were always performed on a per-area basis (i.e., using all components of a given area), rather than on single components. The total explained variance (EV) of a partial decomposition was computed, for each pixel, by computing the variance of $\sigma^2(\sum_{k \in S} A_{x,y,k} C_{k,t})_t$, where S is a subset of components, and where all trials have been concatenated to produce $C_{k,t}$.

The total variance is then summed across all pixels and compared with the variance of the original $F_{x,y,t}$ signal.”

We also now include a new Supplementary Fig. 2 showing examples of spatial distributions for large k values (top 9 components).

(1b) The authors argue for the use of locaMNF by comparing the resulting classification of task-variables to a method working at the level of individual pixels (Supp. Fig. 8). But why is this the meaningful comparison? Would it not make more sense to compare to the performance of a classifier that operates on the entire state-space of pixels (as opposed to individual pixels)? Such a classifier would have access to images of brain activity, like locaMNF, but would not impose any top-down spatial parcellation along area boundaries.

Single-pixel analysis was done to compare with studies that followed a similar local analysis, finding, for example, no evidence of choice-related information in these regions (e.g., Steinmentz et al, *Nature*, 2019; Salkoff et al., *Cerebral Cortex*, 2020; Zatzka-Haas et al., *bioRxiv* 2021; also in the Introduction and Discussion). We concur with the reviewer that a comparison with an entire state-space decoder would be more meaningful if we were to evaluate the superiority of locaNMF against other methods, but this was not within the scope of our analyses. It is indeed conceivable that other system-wide decoders could achieve similar performance; but, as demonstrated in Saxena et al., 2020, the locality constraint in locaNMF adds significantly to the interpretability of the results. The top-down parcellation is justified from previous findings based on widefield and 2-photon imaging (e.g., Minderer et al, *Neuron* 2019; Saxena et al, *Plos Comput Biol* 2020; Abdolrahmani et al, *Cell Rep* 2021), showing significant spatial correlations across large cortical domains in the neural activations associated to behavioral variables. The fact that the seed-area boundaries are not sharp, allowed us to highlight activity patterns that extended across boundaries. We have added these observations to the Discussion, Line 298:

“The imaging methodology and data analysis used in this study facilitated the identification of distributed choice signals encoded by sparse populations of cells⁵⁶. Indeed, sparse encoding in multi-region networks would make choice signals hard to detect with methods that examine decision information independently at each cortical location, whether because of the use of a single electrode (or multi-contact electrode shank)²⁴ or because of imaging data analysis focused on individual locations (pixels) independent from each other^{26,27,57}. We confirmed this observation by reanalyzing our data at the single pixel level: assuming independence between the activations of different pixels, we failed to detect choice information both before and after movement onset. This result may explain why some recent mouse studies have failed to detect choice signals in posterior cortices during similar visually guided tasks²⁴⁻²⁷.

Our choice of seeding-area sizes was chosen to approximately match the spatial correlation length of visual or movement components using smooth penalty boundaries, with sharp boundaries imposing an unnecessary split of components at the border between seeding areas. Small changes in seeding parameters (areas size, centers, smoothness of the boundaries) did not significantly affect the properties of the state axes, suggesting that the spatial correlation length of broadly distributed choice signals approximately matched that of visuomotor components. Additionally, as previously reported⁹⁵, the robustness and stability in spatial profiles across experimental sessions enabled by this seeding method follows from two main

observations: 1) the boundaries between visual areas are identified by gradient flips in field sign maps, but the identification of the boundaries can be imprecise, especially for the smaller higher visual areas (as we observed in our data and as discussed, e.g., ref⁵⁸). Allowing a seeding boundary to extend beyond retinotopic boundaries reduces the dependence of the results on the precise identification of these boundaries. 2) Activity components do not have to follow retinotopic boundaries (e.g., if representing afferent signals from other brain regions or top-down modulations), but these activations are still likely to be spatially correlated. A loose (penalty-based) boundary component allows to capture spatially coherent signals crossing area boundaries.”

(2) State axes

(2a) I found it difficult to evaluate some of the results without a better understanding of the structure of the behavior. How correlated are the various behavioral states that are being used to define state-axes? Correlations between behavioral variables may affect the interpretation of the inferred state axes. In particular, do the small angles between state axes in Fig. 4a reflect a property of the neural representations, or do they reflect strongly correlated behavioral variables?

We thank the referee for pointing this out, as the discussion of correlated variables was missing in the previous submission. Indeed, some of the task variables are highly correlated, as reported in a previous study from our laboratory (Abdolrahmani et al., Cell Rep 2021). For example, wheel and eye movements sometimes occurred near-simultaneously, wheel movements often happened right after stimulus presentation, and wheel movements were also more likely to appear in trials with elevated sustained attention. In view of these correlations, to define the choice axis we only used trials where no other identified movement or saccade took place (Methods, Line 613: “we restricted the analysis to trials in which the first movement occurred at least 0.5 s after stimulus onset and with no saccades 0.5 s before the detected movement”). Similarly, to define the wheel movements and saccades state axes we only used the subset of trials where only one of the two occurred (Methods, Line 594: “Group **A** consisted of trials for which the first movement after stimulus presentation occurred at least 0.5 s after stimulus onset and without any saccade detected in the previous 0.5 s.” and Line 599: “Akin to the definition of wheel movements, Group **A** consisted of trials for which a saccade after stimulus presentation occurred at least 0.5 s after stimulus onset and without any wheel movement detected in the previous 0.5 s. These trials were aligned to the detected saccade time. Group **B** consisted of trials with no saccade detected during the first 5 s after stimulus onset.”. In this revised manuscript, we have included a new analysis demonstrating that significant discriminability of a given state axis is maintained after orthogonalization with other axis (i.e., choice vs attention, movements, and saccades; movements vs saccades and attention; and choice in high and low attention states). This information has been added to the Results, Line 195:

“We verified that the smaller angles observed between some state axes in Fig. 4a were not a direct consequence of correlated behavioral variables by computing their discriminability power after orthogonalization with the other axes (choice vs attention, movements, and saccades; movements vs saccades and attention; and choice in high and low attention). All state axes retained significant discriminability after orthogonalization (Supplementary Fig. 11).”

(2b) State axes are defined by comparing two sets of trials/responses. However, in the definition of the stimulus axis, and potentially of other axes as well, the within-trial timing of the two responses is not matched. For example, in Fig. 2d time points in the no stimulus response occurs earlier in the trial than “time-matched” points in the stimulus response. The corresponding stimulus axis could thus potentially reflect “condition-independent” (i.e. time-dependent) variance (see dPCA) in addition to stimulus-dependent variance. Such variance could affect the angle between state-axes.

The referee is correct, but with the within-trial timing mismatch occurring only for the stimulus state vector. State vectors for choice and attention were always time-matched. Instead, for wheel movements and saccades, in the ‘no-event’ condition, the data was aligned to a random point within the same time interval where an event could occur, which should mitigate the contribution of, for example, a time-dependent component in the response variance. For the stimulus axis we followed a somewhat similar procedure: in our task design the ITI is randomized (3-5 sec), therefore, by choosing time points immediately before the stimulus presentation for the ‘no-stimulus’ condition, time-dependent components in the variability should be significantly averaged-out out by this procedure. However, if mice had learned, for example, an “hazard function” compatible with the ITI randomization (for which we do not have compelling evidence), a time-dependent component may still remain. We have clarified these considerations, as well as improved our description of how the state vectors have been computed in the Methods, Line 587:

“Stimulus: For stimulus state-vectors, we used for the first group (A) trials in the time interval (-0.5 to +0.5 s) centered on the stimulus onset. For group B, since the stimulus was present in all trials, we used the same trials in the preceding time interval (-1 to 0 s) as the no-stimulus condition. Since the preceding ITI was randomized (3-5 sec), time-dependent contributions to the signals should be similar across the two groups. The stimulus condition had on average 2730 trials per animal.”

(2c) Line 525: Stability of state axes. Are the described computations performed separately by area, and stability plots are then averaged across areas? If so, would it not make sense to assess stability at the “whole-brain” level?

Thanks for pointing out this confusing point. We agree that it is better to assess the stability at the whole-brain level, which is indeed what we did in our analysis by using all components. This is now better explained in the Methods, Line 576: *“To assess the stability of state axes, we used our original definition of the time-dependent state axes, that is, using components from all the areas, and a “backward” three-frame averaging window (around 100 ms) and then computed its temporal autocorrelation $C(S(t), S(t'))$. For sensory, movement, and sustained attention state axes we chose the time-independent state axes $S \equiv S(t^*)$, where t^* was chosen from the largest stability cluster (represented by a gray bar in the respective figures). For the state axis of choice, we used the original $S(t)$ to monitor when choice information first appeared and whether its signature was unique.”*

(3) Attention vs choice. Fig. 3j provides a summary of the interaction of choice, task-difficulty, and attention. However, were all aspects of that panel verified in the data? For one, task-difficulty is shown as affecting

responses along the choice axis, but orthogonal to the attention axis, and the choice axis is shown as being identical in the two attention states. But choice axes estimated separately for low and high attention were not identical (line 178), potentially indicating a component of choice that is not maintained across attention states. Also, d' for choice seems larger in the low-attention state (Fig. 3h) compared to high-attention, which is not reflected in panel j? The authors should clarify how the summary panel relates to these findings.

We now clarify that each choice axis was independently defined, based on a different set of trials, and to demonstrate they were significantly maintained across attention states, we performed a new analysis in which we tried to discriminate between left and right trials in one attention state using the axis defined in the other attention state, Supplementary Fig. 10. Regarding the dependence with difficulty, we did observe a clear separation only during high-attention states. The separation during low-attention states was not significant. Regarding the d' for choice being larger in the low-attention state, we have added a new plot: Fig. 3h showing those d' differences are not significant, collectively. Finally, we also agree that the schematic in Fig. 3j was not an accurate summary of the interactions; we have replaced the schematic panel Fig. 3j with one computed using representative trajectories from the experimental data.

(4) RNN modeling. (4a) It is not easy to compare the RNN response to the neural responses, because RNN responses are illustrated with somewhat different figures than the neural responses. For example, in the neural data d' for choice appears to be larger for low vs high attention (fig. 3h,i). That does not appear to be the case in the RNN (Fig. 5g), although a direct comparison is difficult because the plots are not the same. The authors should simplify the comparison, and clarify what aspects of the neural responses are captured by the RNN, and which are not.

We have updated most of the panels in Figure 5 (b to j) to allow for a better comparison between neural and model data. We have also performed new, extensive simulations, with a new parameter set; Methods: Line 655, using $N=50$ neurons instead of $N=128$, and Line 666: “We used a constant $\lambda = 0.2$ and $\alpha = 2/90$ for low attention and $\alpha = 5/90$ for high attention. Lapse rate was chosen to match that of a representative animal. Slopes were chosen to match average performance across difficulties of a representative animal (67% and 80% for low and high attention respectively; sampled from a uniform trial difficulty distribution).” This new parameter set better captured the discriminability values obtained from experimental data. The comparison between experiment and model should now be significantly clearer. We have also updated the Results to clarify which aspects of the neural response are captured by the RNN, Line 247:

“Performance levels and differences between high and low attention states were chosen to match experimental values. We then analyzed the internal dynamics of the network by computing choice and attention axes from RNNs unit responses, as we did for the neural data with locaNMF components. In the RNNs the choice axis identified a decision variable that represented L/R decisions as separate trajectories in a low-dimensional embedding space (Fig. 5b-d). Furthermore, the separation between L/R trajectories was modulated by attention and task difficulty, with larger separations in easy trials and high attention (Fig. 5e-g). This separation did not depend on absolute orientations, as expected from the RNN having learned this invariance. Attentional modulations maintained an invariant representational geometry of the decision variable across

the embedding space, that is, the choice axis remained stable with attention and task difficulty (Fig. 5h-i). This was consistent with what observed in the neural data, where choice and attention axes were near-orthogonal with each other (Fig. 4a). Although the model was trained only with a subset of 13 difficulties and two attention states, it was able to generalize to any difficulty level and range of attention within the trained boundaries (Fig. 5j).

In summary, the similarity of the representations between the RNN and neural dynamics, suggests that the contextually modulated choice signals observed in locaNMF components represented the decision-making computations underlying this task, as learned by the RNN when using the same behavioral output of the animals.”

Regarding the d' differences in the data as in Fig. 3h-i, we believe this point has now been addressed in our reply to the previous question.

(4b) Line 144: the authors emphasize that the RNN generates probabilistic choices even though it was trained on binary choices. But my understanding of the methods was that the output neurons implement a (probabilistic) softmax output function? If so, is it not a given that the choices of the RNN will describe a smooth psychometric curve?

We agree with the referee, that is indeed the case, and that the previous formulation was rather imprecise. The simple message wanted to convey is that by training with the (probabilistic) animal choices, rather than with the task rule (as it is typically done in similar simulation studies), the output neurons correctly learned to reproduce the psychometric responses of the animal. We have updated the text, Line 242: “*As a result, RNNs learned to produce L/R binary choices from an internal representation that followed a psychometric probability function based on the absolute difference between the two inputs*”, and Fig 5a. accordingly.

Minor point

The introduction cites papers across a range of species (primates + rodents), areas (from early sensory to late association and movement), tasks, and methods. While I appreciate the effort at being scholarly and the attempt at linking the presented research to the whole field, some of the references are very brief and compressed. Overall, I found it hard to follow what the authors meant to say, i.e. why a particular reference is used. In essence the authors are providing a review in a few paragraphs, which has the disadvantage of somewhat hiding/obscuring the goals of this study.

Based on the referee’s feedback we have substantially simplified and streamlined the Introduction.

Reviewer #2

Several studies have shown that the representations of choice information can be found the occipital, parietal, and frontal cortices, but determining whether these representations across areas are comparable, and separating these choice signals from stimulus and motor information, has been challenging. To investigate choice representation, mice in this study are trained to perform a challenging 2AFC orientation discrimination task during mesoscale wide-field imaging of the posterior cortex. The study employs a tensor decomposition method, locaNMF, to identify unique spatiotemporal components in the imaging data. They then apply an additional dimensionality reduction step, revealing the underlying representation of stimulus, movement and choice information in each area. With support from a recurrent-neural network trained on the animals' behavioral data, this study demonstrates that the posterior cortex contains a multi-area decision variable that is modulated by task and behavioral variables. This exciting study provides new insight into the representation of choice across cortical areas and provides an experimental and analytical framework for further investigations. Overall, the study is clearly presented, and the authors are commended for their multiple experimental and analytical controls throughout. I only have a few suggestions for additional improvement to the transparency of an already excellent study.

We thank the reviewer for the appreciation of our work.

1. Figure 1d. The authors should consider using 'Probability right choice' to match with convention, and 'Ignore (%)' instead of 'Time out (%)' since the cause of time out isn't immediately clear.

We have changed the figure panels accordingly.

2. Figure 1e. The authors describe alignment of the multiple higher visual areas with the Allen Institute Atlas using the retinotopic field-sign maps. The authors should provide an example field sign map to show the level of confidence in area assignment.

Following the reviewer's suggestion, we have included an example field-sign map in Supplementary Fig. 1, together with the alignment of our retinotopic boundaries to the Allen atlas. Furthermore, we now mention that the confidence in area identification based on the field-sign maps was also demonstrated in a previous work, Line 69:

"We aligned all imaging sessions according to the Allen Common Coordinate Framework³⁹ (Fig. 2a), and seeded the initial spatial decomposition using 10 large regions centered on retinotopically identified areas (based on field sign maps, Supplementary Fig. 1a, see also ref²⁰) that extended significantly beyond area boundaries (Supplementary Fig. 1b)."

3. Supplementary Figure 1a. The authors provide penalty maps that describe seeding areas and area boundaries for locaNMF decomposition. What benefit is gained from having boundaries that extend significantly beyond areas? How much does the seeding impact the resulting special components? The authors should address this with additional commentary.

Indeed, these important considerations were not thoroughly discussed. We now explain two main benefits of having boundaries extending significantly beyond the areas:

1) Boundaries between visual areas are identified by gradient flips in field sign maps, but the identification of the boundaries can be imprecise, especially for the smaller higher visual areas (as we observed in our data and as discussed, e.g., in Zhuang et al., eLife 2017). Allowing a seeding boundary to extend beyond retinotopic boundaries reduces the dependence of the results on the precise identification of these boundaries.

2) Activity components do not have to follow retinotopic boundaries (e.g., if representing afferent signals from other brain regions or top-down modulations), but these activations are still likely to be spatially correlated. A loose (penalty-based) boundary component allows to capture spatially coherent signals crossing area boundaries.

As mentioned in reference to a similar question raised by Reviewer #1, the spatial profile of the components is quite robust to slight variations in the seeding regions and their shape. However, a finer, or coarser seeding structure would indeed change the overall structure of the components. For example, if one was interested in identifying the position of different visual stimuli, one could subdivide V1 in several subregions. Based on our analyses, we believe that a “loose” seeding on retinotopic areas achieves a good compromise between spatial localization and interpretability of the components for our task. We have added these considerations in the Discussion, Line 298:

“The imaging methodology and data analysis used in this study facilitated the identification of distributed choice signals encoded by sparse populations of cells⁵⁶. Indeed, sparse encoding in multi-region networks would make choice signals hard to detect with methods that examine decision information independently at each cortical location, whether because of the use of a single electrode (or multi-contact electrode shank)²⁴ or because of imaging data analysis focused on individual locations (pixels) independent from each other^{26,27,57}. We confirmed this observation by reanalyzing our data at the single pixel level: assuming independence between the activations of different pixels, we failed to detect choice information both before and after movement onset. This result may explain why some recent mouse studies have failed to detect choice signals in posterior cortices during similar visually guided tasks²⁴⁻²⁷.

Our choice of seeding-area sizes was chosen to approximately match the spatial correlation length of visual or movement components using smooth penalty boundaries, with sharp boundaries imposing an unnecessary split of components at the border between seeding areas. Small changes in seeding parameters (areas size, centers, smoothness of the boundaries) did not significantly affect the properties of the state axes, suggesting that the spatial correlation length of broadly distributed choice signals approximately matched that of visuomotor components. Additionally, as previously reported⁹⁵, the robustness and stability in spatial profiles across experimental sessions enabled by this seeding method follows from two main observations: 1) the boundaries between visual areas are identified by gradient flips in field sign maps, but the identification of the boundaries can be imprecise, especially for the smaller higher visual areas (as we observed in our data and as discussed, e.g., ref⁵⁸). Allowing a seeding boundary to extend beyond retinotopic boundaries reduces the dependence of the results on the precise identification of these boundaries. 2) Activity components do not have to follow retinotopic boundaries (e.g., if representing afferent signals from other brain regions or top-down modulations), but these activations are still likely to be spatially correlated. A loose (penalty-based) boundary component allows to capture spatially coherent signals crossing area boundaries.”

4. Supplementary Figure 1c – On lines 95-97 the authors describe that each main component of a seeding area contributes, on average, 9.6 % of the total explained variance. However, this figure shows that the first component contributes ~45% explained variance. Why is this initial component not addressed? In addition, what conclusions should the reader reach by the comparison of the SVD and locaNMF methods? The figure shows SVD achieving larger explained variance (~80%) than locaNMF with fewer components. Yet, the implication is that locaNMF is a better approach. The authors should unpack this comparison of SVD and locaNMF so the reader can better understand the determining factors in the decision to utilize locaNMF.

We thank the referee for these observations which have highlighted some confusing phrasing. On average, the main component of each seeding area contributes 9.6 %, however, the contributions from the main component of each area are highly heterogeneous. As the referee pointed out, one component (from V1) already accounted for 45% of the variance. We have rewritten the text to better represent this effect, Line 79: *“The largest locaNMF component of each cortical area provided significant explanatory power, together contributing to 96 % of the total explained variance (Supplementary Fig. 1c). These contributions being highly heterogeneous, with the largest component of V1 accounting for 45% of the total variance.”* We have also expanded on the locaNMF and SVD comparison in the Discussion, Line 279:

“We achieved these results by combining two powerful methods for the analysis of population responses: locaNMF and activity-mode analysis. LocaNMF reduced the dimensionality of the neural data while retaining spatial information, that would have been lost with traditional dimensionality reduction methods (e.g., SVD, NMF). Traditional decomposition methods could also be used to reduce data dimensionality, but the loss of spatial localization would greatly reduce the interpretability power of the components, and the components would also be less robust to session-to-session variability, as previously reported³⁵. Furthermore, the state space representation allowed further reduction of dimensionality by aligning the dynamics along task and decision-relevant dimensions. This latter step took place within an interpretable linear framework, where the angle between the state axes as well as d' values could be directly linked to the linear discriminability of the underlying variables.”

5. Figure 2d-g. To provide the readers with an intuition regarding statistical power, the authors should provide details regarding the trial numbers for each particular state axis (i.e. high attention vs low attention).

We have included these numbers in the Methods for all task-related state axes. For the particular case of choice and attention, Line 614:

“This condition had on average 657 trials per animal. When splitting the trials based on attention, the number of available trials reduced to, on average, 200 trials (high and low attention trials only accounted for 33% of the original trials each). When trial difficulty was also included, the number of available trials reduced to 70 on average (each difficulty split further decreased the number of available trials on each condition by slightly more than one half).”

6. Figure 2g. This figure describes sustained attention based on pupil area changes. How are reward and other known factors that influence pupil size accounted for in this description? The authors should provide

additional discussion addressing the illustrated dynamic and the impact of other factors that influence pupil size.

We have included a new discussion paragraph on the components that affected pupil area changes in our task, Line 356:

“Attention-mediated modulations were identified via an analysis of pupil area changes, known to reflect variability in internal states of the animals due to changes in engagement and vigilance during the task, with “sustained” attention” referring to a broad spectrum of these goal-directed internal states^{20,47,48,68}. In the time window of our analyses, the main event triggering changes in pupil area was the stimulus onset. However, at trial end a high-contrast checkerboard (false alarms) and water reward (hits) could have also triggered pupil area changes.”

7. Figure 3a. The projection of the right choice is similar to that of the left choice briefly after movement onset, before a reversal. Does this reflect an intrinsic choice bias, or an inability to completely collapse along this state axis? The authors should describe this observation and how dynamics emerge.

The trend shown in Fig. 3a is for one example animal, and it is likely due to an overall increase in cortical activity. The data projected on the left-right difference-axis are not mean-centered or z-scored at each time point, thus overall activity changes at specific times can cause co-fluctuations in projected values in both left and right projected trials – in this example shortly after the stimulus onset. However, as shown in the d' curve these co-fluctuations do not affect the discriminability d' , which increases smoothly over time, a trend observed consistently across animals. We have added these considerations in the Methods, Line 565: *“Because there is no mean-centering in time of the components, i.e., $\langle C^i(t_j) \rangle_i \neq 0$, activity changes across all areas can cause co-fluctuations in the projected trajectories.”*

8. Figure 3f and 4d. It is not clear how global d' is calculated and what is its relationship to the local d' in each area (it seems like it is sublinear). Similarly, I am confused by the SDI index. It seems like the assumption is that if the max area d' is high relative to the global, then this suggests that it can account for most of the total d' , and therefore it is not evenly distributed. However, this makes assumptions about the linearity of d' summing across areas to create the global d' .

The referee is correct in this interpretation. The global d' is computed by using all components from all areas at once, and the local d' by only using the components of a given area. We have improved the related descriptions in the Methods, Line 637:

“The SDI for a given state axis was computed as $SDI (\%) = \left(\frac{d'_{global}}{\max(d'_i)} - 1 \right) \cdot 100$ where d'_i refers to the area-specific state axes d' scores (i.e., a state axes defined using only the weights of i -th area), and d'_{global} refers to the discriminability of the original state axis (i.e., the state axis defined using all components). Since the original state axis uses all components across the areas, d'_{global} is an upper bound of d'_i . Similarly, each d'_i contributes sublinearly to d'_{global} , since correlated activity across areas would not result in higher discriminability. Hence, SDI measures how much the discriminability can increase when using all areas vs just the one with the largest d' . The SDI plays a similar role to deviance-explained in statistical models, as it measures the relative

*contribution of a small model (single area) vs the full model (all areas). In this definition, SDI = 0% if only 1 area contributes and $(N-1)*100$ % if all N areas contribute equally and independently.”*

9. Figure 4b – The authors use hierarchical clustering to illustrate the relationship between state axes. The authors note that early and late choice are orthogonal from each other, but make stronger statements about their orthogonality from the other variables. Notably, early and late choice are more different from each other than any of the other variables are different from each other. This result could use some more explanation: what are the major differences between choice early and choice late?

Differences between early and late choice come from the spatial distribution of their components. Although the discriminability power of early choice was small (Fig. 3b), it was significant ($p < 0.001$), and its spatial contribution was less localized than late choice (from the SDI index, Fig. 4d), with each cortical region contributing with similar values (Fig. 4e). On the other hand, choice late was more localized (higher SDI), with bigger contributions on the ventral stream (relative to the dorsal and posterior broad areas, Fig. 4e). We have rewritten the results section related to axes orthogonality (Line 188) and added a new paragraph stating the differences between choice pre and post movement (Line 226): *“These results highlighted the major differences between choice state axes defined pre and post movement. Choice pre-movement was less localized (lower SDI), with each cortical region contributing similarly. On the other hand, choice post-movement was more localized (higher SDI), with larger contributions on the ventral stream (relative to dorsal and posterior broad areas, Fig. 4e).”*

10. Figure 5f & g- There appears to be a discrepancy between these two figures regarding d' amplitudes and timing. Panel g separates the hard and easy trials into high attention and low attention, but after this separation the max d' is lower (~ 5 compared to ~ 7). In addition, the timing of the change in d' in panel g is shifted to the left as compared to panel f. The authors should describe the source of this discrepancy and any implications that it may have on the interpretation of these results.

We thank the referee for pointing this out. The timing issue was due to a 1-frame offset in the labeling, but the timing of d' change in both figures was indeed the same. Regarding the difference in max d' , the large confidence intervals and the use of different y-scales made indeed a careful comparison difficult. Because of this, and for the discrepancy in d' values relative to the neural data (as noted by another referee), we have improved on the simulations by (1) training the networks with reference psychometric curves having the same average performance as the one of a representative animal (in the previous version, the difference between the high and low attention psychometric curves was much larger than in the experiments). (2) Using fewer units in the RNN module to avoid overtraining (Methods, Line 654, 50 instead of 128), and ensuring all the models were trained successfully, i.e., comparable loss in training and test sets; this is because in some instances, random initialization of the parameters resulted in the model failing to converge. These changes collectively have allowed for a more quantitative comparison with the neural data as in the new Figure 5.

11. Supplementary figures 5 and 7 have typos in the captions.

a. Supp 5 - The caption for panel “c” is labeled “d”.

b. Supp 7 – The caption for panel “b” is “As in H, but after movement onset” when it should be “As in a, but after movement onset”.

We thank the referee for noticing these typos and oversights which have been corrected.

Reviewer #3

The authors recorded wide-field calcium imaging data in mice during a perceptual discrimination task. The task consisted of turning a wheel to indicate which of 2 oriented gratings were most similar to a target orientation. The authors used localNMF to decompose data into spatially- dominated components and then conducted a decoding analysis to establish encoding axes for a variety of task variables.

The experiments were impressively successful and I appreciate the willingness to use mice as a model animal for studying decision making. The analysis was thorough and while most of the authors' claims were substantiated by the data, the authors made some questionable assertions and confusing choices regarding the presentation of the results that I believe can to be addressed in revision. The following are specific points that should be addressed.

We thank the reviewer for the appreciation of our work.

- The authors used localNMF to decompose the wide-field 2P imaging in to pixels associated with particular regions. I wonder, however, what specifically is gain by forcing the weights to correspond to specific regions. Why not use the global encoding axes and ask post-hoc whether the encoding regions correspond to anatomically-defined regions?

We were not sure whether the reviewer was referring to what we called “global state axis” (Methods, Ln 560), so apologies if we the following reply does not entirely address the reviewer’s concern. The main advantage of using the described approach follows a similar reasoning outlined in the LocaNMF study by Saxena et al., (Plos Comp. Biol. 2020), which is basically a substantial gain in “interpretability” of what the obtained weights represent while keeping the number of weights manageable and their spatial signature localized. Interpretability here relates to the goal of extracting mixed signals and reference them (consistently) to well-defined regions, while adapting to anatomical differences between animals and accounting for the possible “mixing” of multiple signals within the same region(s). As demonstrated in Saxena et al., methods such as NMF with atlas initialization (possibly what the reviewer was referring to) produce less consistent localization even across sessions for the same animal. The spatial localization of LocaNMF each component is still allowed to shift slightly from animal to animal, but remains largely consistent within and across animals, while at the same time overcoming significant limitations to ROI based methods (which discard information outside the ROI or cannot demix overlapping signals within the ROI) PCA/SVD methods with typically delocalized components. Note also that the definition of seeding areas with an exponentially decaying penalty function with distance still allows to pick up strong cohesive signal that may span across regions.

We have now included these considerations in the Discussion, new methodological relevance section, Line 279:

“We achieved these results by combining two powerful methods for the analysis of population responses: locaNMF and activity-mode analysis. LocaNMF reduced the dimensionality of the neural data while retaining spatial information, that would have been lost with traditional

dimensionality reduction methods (e.g., SVD, NMF). Traditional decomposition methods could also be used to reduce data dimensionality, but the loss of spatial localization would greatly reduce the interpretability power of the components, and the components would also be less robust to session-to-session variability, as previously reported³⁵. Furthermore, the state space representation allowed further reduction of dimensionality by aligning the dynamics along task and decision-relevant dimensions. This latter step took place within an interpretable linear framework, where the angle between the state axes as well as d' values could be directly linked to the linear discriminability of the underlying variables.

Dimensionality reduction via trial and spatial averaging permits the robust isolation of state axes, however, it can also collapse sub-spaces over which important aspects of the dynamics might evolve. For instance, we found that pre- and post-movement decision axes had large angular separations, defining a sub-space orthogonal to that of movement signals. We also found that the dimensionality of choice signals was further enriched by attentional modulations, which produced an isomorphic shift of choice representations. It is conceivable that extra dimensions linked to more subtle aspects of the decision-making process exist but have been “collapsed” by averaging signals across trials. Future investigations, for example based on probabilistic low-rank dimensionality reduction methods⁵⁵, might be able to uncover these hidden dimensions.

The imaging methodology and data analysis used in this study facilitated the identification of distributed choice signals encoded by sparse populations of cells⁵⁶. Indeed, sparse encoding in multi-region networks would make choice signals hard to detect with methods that examine decision information independently at each cortical location, whether because of the use of a single electrode (or multi-contact electrode shank)²⁴ or because of imaging data analysis focused on individual locations (pixels) independent from each other^{26,27,57}. We confirmed this observation by reanalyzing our data at the single pixel level: assuming independence between the activations of different pixels, we failed to detect choice information both before and after movement onset. This result may explain why some recent mouse studies have failed to detect choice signals in posterior cortices during similar visually guided tasks²⁴⁻²⁷.”

- Figure 3c depicts the cross-correlation between choice axes estimated from different time windows. As the authors pointed out, the 2 distinct regions suggest that there are 2 stable choice axes. They schematically depict this in Figure 4c. However, they could have simply plotted the projection onto each of these axes and demonstrated the plausibility of their hypothesis, rather than presenting their schematic. A similar visualization could have been done for the movement axes. I would have found this demonstration to be far more convincing.

We agree with these considerations and have changed the panel accordingly (See Fig 4c), see also our reply to the related comment below.

- Regarding statements starting on line 201, Figure 4c, The authors claim that the choice and movement axes are always orthogonal to each other but it does not appear to be the case based on the inset. In particular, it appears that prior to movement onset there is some non-zero correlation between axes. Could the authors please explain this?

The average angle from the cosine distance similarity analysis, before and after movement onset, overlaps with the 95% confidence interval of the null model (shaded area, 79 ° to 90 °) before ($77 \pm 3^\circ$ at $t=-0.5$ s; p value = 0.25, one sided t-test against 79 ° lower bound) and after movement onset ($80 \pm 3^\circ$ at $t=0.5$ s; ; p value = 0.73, one sided t-test against 79 ° lower bound); the null model being a randomization (resampling with replacement) of the indices of one of the axes. Hence, we confidently report no significant correlations between choice and movement axes. However, we agree with the referee that this observation does not imply that these axes are “always orthogonal” with each other, thus we have changed the text accordingly, Line 187:

“Irrespective of the time period, choice was nearly orthogonal to the movement axes (Fig. 4c), with no significant differences when comparing with a null model with orthogonal axes; both before ($77 \pm 3^\circ$ at $t=-0.5$ s; p value = 0.25, one sided t-test against 79 ° null model lower bound) and after movement onset ($80 \pm 3^\circ$ at $t=0.5$ s; ; p value = 0.73, one sided t-test against 79 ° null model lower bound);. When transitioning from the pre- to the post-movement period, d' values never collapsed to zero (Fig. 3b), suggesting a rotation of the choice axis while preserving the orthogonality between choice and movement axes. This can be understood as a rotating state axis for choice in a multi-dimensional choice sub-space, that remained orthogonal to a similarly defined movement subspace.”

Given the observed values, we now restrict our claims to the lack of statistically significant angular dependence between the two variables, both before and after movement onset.

- Figure 3j is a schematic illustration based on the authors’ description of encoding geometry in the data and Figure 5h presents this same phenomenon for the data obtained from the trained RNN. It’s unclear to me why the authors (again) did not choose to plot the actual data in this way, which seems to me to be entirely possible considering they did so for the RNN, rather than depicting a schematic representation.

We agree with the referee and have changed both panels using experimental data.

- Line 178 “Choice axes independently defined in low- and high-attention states were highly correlated (Pearson’s $r = 0.72 \pm 0.03$), indicating they reflected a congruent underlying decisional process. “However, $r = 0.72$, while high by some standards, is hardly the same. Can the authors comment on differences in the choice axes between states and explain why these are statistically significantly different?”

We agree with the referee that “high” or “low” are only qualitative connotations for correlation values. We now clarify that, each choice axis was independently defined, based on a different set of trials, and to demonstrate they were significantly maintained across attention states, we performed a new analysis in which we tried to discriminate between left and right trials in one attention state using the axis defined in the other attention state. This procedure maintained a significant discriminability power: Supplementary Fig. 11 and Line 196:

“We verified that the smaller angles observed between some state axes in Fig. 4a were not a direct consequence of correlated behavioral variables by computing their discriminability power after orthogonalization with the other axes (choice vs attention, movements, and saccades;

movements vs saccades and attention; and choice in high and low attention). All state axes retained significant discriminability after orthogonalization (Supplementary Fig. 11)."

- Starting on line 260: representational similarity... I am unconvinced that these assertions follow from the evidence provided. First, what properties of the representations found in the RNN should convince us that the mechanism employed by the RNN is the same as that employed by the mice? Why does the lack of neural data lend credibility to the RNN analysis? I am personally left more skeptical by this feature than less.

We have significantly changed the text in the referenced concluding paragraph. We now clarify that it is not the case that the lack of neural data lends credibility to the RNN analysis per-se. Rather, we observe that a minimal computational model, trained only on behavioral inputs and outputs, evolves an internal low-dimensional representation that recapitulates the main features observed on the neural representations. Because the RNN computations are known, the similarity between the solutions found by the RNN and the representations found in the biological network suggest the latter also reflect computations akin to those learned by the RNN from behavioral data. This inferential logic closely follows similar narratives introduced in previous studies; a well-cited example being the work by Mante and Sussillo (Nature 2013): they trained an RNN on behavioral data (as we did) finding dynamical representation bearing similarities to neural ones (as we did), letting them to conclude neural representations reflected computations similarly learned by the RNN from the behavioral data. We hope the revised text will convey this message more clearly.

- The wheel movement axis (Supplementary Fig. 2b) does seem to show a dramatic shift at the time of movement onset but the cross-correlograms suggest a constantly, albeit slowly, hanging axis in both the pre- and post- movement periods. This undermines the authors claim that there are 2 movement axes that shift around the time of movement onset. Can the authors please explain?

On the cross-correlogram of Supplementary Fig. 2b we show the correlation between state axes in different time intervals, from zero-time separation (the diagonal), up to 2 s (at the "corners"). Due to continuous changes in overall activity over time, the axis found at any given time point will necessarily show "some" decrease in correlations with axes computed at progressively larger time distances ("slow drift"). However, correlation values for axes computed at nearby time-points are typically large (close to 1). As the referee pointed out, the cross-correlograms also show a dramatic shift around the movement time, where any axis defined post-movement is essentially orthogonal to any axis defined pre-movement (1st and 3rd blue quadrants in Supplementary Fig. 2b), hence our statement of there being two movement axes that shift around movement onset.

Rebuttal Figure 1: time-slices of Supp. Fig. 2b and hierarchical clustering of movement axes, with clusters for pre- and post-movement time.

In this figure we show in panel (a) a cross-section of the cross-correlogram from Supp Fig 2b for two specific state axes (pre and post movement, defined at ~ 200 ms before and after movement onset, i.e., at times when correlation values $r = 1$ in the blue and orange curve, respectively). Note that correlation values remain large within the pre and post movement periods, with correlations quickly dropping to near zero values in the “other” time interval (positive times for the blue line, and negative for the orange; lines and shaded regions are mean and s.e. of the cross-correlogram averaged across $n = 7$ animals). This observation is reflected in panel (b) where we performed hierarchical clustering (average linkage with correlation metric) of the movement state axis defined pre and post movement for a given animal. The state axes cluster in two large groups corresponding to pre and post movement onset (clearly seen by the change in sign). Only the first two frames post movement belong to the pre movement cluster, and this is due to the moving window used to average the data when defining the state vectors (as detailed in the Methods). However, we agree all these considerations were not clearly explained; therefore, we have rephrased the related statements as follows, Line 187:

”Irrespective of the time period, choice was nearly orthogonal to the movement axes (Fig. 4c), with no significant differences when comparing with a null model with orthogonal axes; both before ($77 \pm 3^\circ$ at $t=-0.5$ s; p value = 0.25, one sided t -test against 79° null model lower bound) and after movement onset ($80 \pm 3^\circ$ at $t=0.5$ s; ; p value = 0.73, one sided t -test against 79° null model lower bound);. When transitioning from the pre- to the post-movement period, choice d' values never collapsed to zero (Fig. 3b), suggesting a rotation of the choice axis while preserving the orthogonality between choice and movement axes. This can be understood as a rotating state axis for choice in a multi-dimensional choice sub-space, that remained orthogonal to a similarly defined movement subspace.”

- Line 344. The authors’ comment about “forcing” a shift on other task variables doesn’t make sense to me. An encoding axis can shift and remain orthogonal to the other task variable axes without any change in the other axes. This is particularly easy in high dimensions.

The reviewer is correct and have reworded the statement accordingly, Line 380:

“These large activations occupy the main dimensions of variability²¹, enabling a representation shift while keeping movement representations separate from other variables, that is, in near-orthogonal subspaces.”

- The spatial averaging imposed by localNMF decreases the effective degrees of freedom of the data. Averaging across trials to estimate state axes again reduces the effective degrees of freedom. Can the authors comment on how the effective degrees of freedom of the data could impact the spatio-temporal analysis that they have depicted?

We were not entirely sure about this question, which we interpreted as how our analyses and methodology might have impacted the effective degrees of freedom—rather than the opposite (how the effective degrees of freedom impacted our analysis). Our approach certainly decreases the effective degrees of freedom in the data. However, to begin with, widefield signals are characterized by a relatively large spatial-correlation length (of the order of 100 μm rather than few μm), with spatial averaging used in our study simply reducing trial-to-trial variability at this resolution, rather than “obscuring” spatial features possibly existing at a finer spatial scale. In time, we have temporally aligned data to the events of interest, that is, stimulus or movement onset times. Therefore, trial averaging is “smearing out” any component that is not consistently time-locked to these events. We believe this is the case, for example, for decision signals as in Fig. 3b, with significantly positive (and ramping-up) d' values before movement onset. We have added additional commentary on this topic in the Discussion, Line 430:

“Our study focused on features of choice signals that were stable and consistent relative to the temporal structure imposed by our task design. However, it is very likely that other task-uninstructed components (e.g., motor, attentional, decisional) might exist, and more in general, components that do not bear a systematic temporal relation with the trial structure, and therefore characterized by a large trial-to-trial timing variability within and across trials. Our temporal alignment and trial-averaging procedure would average-out these components, thus reducing the effective degrees of freedom of the representations.”

- Line 355: “representational similarity analysis...” The authors here reference Mante et al. (2013) (reference 81). However, I can find no reference to representational similarity analysis (RSA) in this paper. I should point out that RSA is often referred to as a formal analysis method. Perhaps there is some colloquial sense in which the authors are using this term? If so, I suggest the authors choose a different description of this family of analyses.

We agree with the referee and have changed the text accordingly, Line 390:

“Recurrent state-space models, including RNNs, have been previously used in mechanistic investigations of decision-making processes^{77,78}. Moreover, analyses of the similarity of the state-space representations in RNNs and neural responses has been successfully used to infer underlying computations⁷⁸. Here, we adopted a similar approach, but with three main distinguishing features.”

REVIEWER COMMENTS

Reviewer #1 (Remarks to the Author):

The authors have addressed my questions and concerns.

Reviewer #2 (Remarks to the Author):

The authors have greatly improved their revised manuscript. However, there are still a few issues that were not satisfactorily addressed.

1. I am still unsatisfied that the measure of maximum change in pupil diameter between windows before and after stimulus presentation (which includes reward delivery) uniquely reflects sustained attention as the authors purport. The addition of a sentence in the discussion suggesting that it may also reflect the outcome is too little and too late given the weight that the study puts on the measure in figures 2-5. The authors should demonstrate that there is some orthogonality (understood that it might not be complete) between their measure of sustained attention and trial outcome. In addition, perhaps it makes sense to see if the result holds if the authors restrict their pupil analysis window to a reflect stimulus-driven changes in pupil diameter (which have been shown to correlate with attentional state), or simply the baseline pupil diameter preceding the trial.

2. I am also still not clear on how to think about the difference between the choice signals in the pre- vs post-movement period. The authors have added some description that these choice signals have differential localization- pre being global and post being localized to the ventral stream (area L). However, little is done to describe whether these signals in fact reflect the same information or different information- and what that might be. In addition, there is some lack of clarity on the separation of these two time windows in the discussion. For instance, effort is given in figure 3 to demonstrate that there is choice information preceding the movement, and this is used to argue that this is therefore distinct from the left/right movement per se (lines 337-338). But this only applies to the pre-choice window. It is also notable that the pre-window is relatively weak (low d-prime) while the post window is much stronger. More needs to be done to address the conceptual difference between these choice signals, and why they are orthogonal.

3. The authors have added trial numbers which are extremely helpful- and seem to be well above the threshold for significance. However, only a single number was given per A/B condition- and while some A/B conditions should be matched (e.g. sustained attention and stimulus), others are likely not (e.g. wheel movement and saccade) and so should have two numbers.

The authors have greatly improved their revised manuscript. However, there are still a few issues that were not satisfactorily addressed.

We are grateful for the appreciation of the hard work in revising the manuscript and we thank the reviewer for the very helpful comments.

1. I am still unsatisfied that the measure of maximum change in pupil diameter between windows before and after stimulus presentation (which includes reward delivery) uniquely reflects sustained attention as the authors purport. The addition of a sentence in the discussion suggesting that it may also reflect the outcome is too little and too late given the weight that the study puts on the measure in figures 2-5. The authors should demonstrate that there is some orthogonality (understood that it might not be complete) between their measure of sustained attention and trial outcome. In addition, perhaps it makes sense to see if the result holds if the authors restrict their pupil analysis window to a reflect stimulus-driven changes in pupil diameter (which have been shown to correlate with attentional state), or simply the baseline pupil diameter preceding the trial.

We acknowledge the concern raised by the referee about our measure of sustained attention, also in reference to the last sentence added to the discussion which was not precisely formulated thus generating some confusion. The previous text regarding the definition of sustained attention only mentioned the maximum area change after the stimulus onset, without specifying the time window, which was during the open-loop period, thus within 1.5 s after stimulus presentation (the same criteria used in previous work: Abdolrahmani et al, Cell Rep., 2021). The analysis of the pupil dilation indeed measures the change in pupil area due to the stimulus presentation in the open-loop period before reward delivery (actions resulting in a reward or punishment only happen in the closed loop period). We have now corrected the parts in the Results section describing how the “sustained attention axis” was computed (Ln 122):

“Based on the pupil area increase within a short time window after stimulus presentation (open loop, Methods), we defined a state axis that discriminated between states of high and low sustained attention (Fig. 2g).”

And in the Methods section (Ln 637):

*“Sustained attention was measured by changes in pupil area during the stimulus presentation. We computed pupil area changes (pA) as the difference between the maximum pupil area during the open-loop period (i.e., the 1.5s window after stimulus onset) and the average area 1 s before stimulus onset. We labeled as “high sustained attention” trials (group **A**) those in the top 33rd percentile of the pA distribution and as “low sustained attention” trials (group **B**) those in the bottom 33rd percentile. Groups **A** and **B** were always balanced by definition.”*

Similarly, the baseline value of pupil dilation in each trial is determined during a 1 s epoch before stimulus presentation, which, given the distribution of intertrial intervals, is far enough from the previous reward or punishment to have any transient effect on pupil area. Therefore, what the referee suggested is indeed what we did in our

analysis, that is, defining pupil area changes in a time window that best captures stimulus-driven changes in pupil dilation.

However, to quantitatively address the reviewer's concern about a possible relationship between trial outcome and pupil changes around stimulus onset, we computed a new pair of attention axes: one using only correct trials (when water reward was delivered) and the other with only incorrect trials. We observed that the angle between these two vectors was "small" (22 deg on average), slightly larger than the expected value for parallel vectors given the variability in the data, that is, the average angle between the same state axis defined using different folds of the cross-validation procedure (approximately 13 deg on average - new Supp. Fig. 8a). These angles were far from being orthogonal (dashed line), highlighting that the definition of the sustained attention axis was robust relative to trial outcome. Additionally, we also measured the Pearson correlation values between the two state axes, with a correlation of $r = 0.92$, also highlighting their similarity (Supp Fig. 8b).

Furthermore, we examined whether the discriminability of states of sustained attention was robust relative to a "swap" between the two axes. That is, the discriminability between high and low attention states on correct trials was maintained if we used projections onto the axis defined using incorrect trials, and vice-versa (Supp. Fig. 8c). Although independence is hard to prove explicitly, these results suggest that the combination of locaNMF components that best capture attentional information are rather independent of the trial outcome.

We have edited the manuscript to reflect these new analyses and considerations in the results section (Ln 129):

"The attention state axis orientation was stable relative to the trial outcome (correct or incorrect); the angle between the state axes for sustained attention defined using either correct or incorrect trials was $23^\circ \pm 2^\circ$, slightly larger than the expected value for parallel vectors given the variability in the data, that is, the average angle between the same state axis defined using different folds of the cross-validation procedure ($\sim 13^\circ$ deg on average, Supplementary Fig. 8c). The d' values obtained when discriminating attention states from correct trials using their projections onto the state axis defined with incorrect trials and vice versa were comparable ($d' = 1.17 \pm 0.07$ and $d' = 1.2 \pm 0.1$ respectively)."

And in the discussion section (Ln 383):

"Attention-mediated modulations were identified via an analysis of pupil area changes around the time of stimulus onset, known to reflect variability in internal states of the animals due to changes in engagement and vigilance during the task, with "sustained attention" referring to a broad spectrum of these goal-directed internal states^{20,47,48,69}. In the time window of our analyses, the main event triggering changes in pupil area was the stimulus onset. The variability in pupil area changes correlated with task performance and response times (Supplementary Fig. 7a, b) and could also be associated with "sustained" changes in cortical activity, as demonstrated by the presence of a stable attention axis throughout trial time (Supplementary Fig. 3d). The axes stability was not linked to trial outcome, which would have been the case if the changes in pupil area were related, for example, to reward delivery or other task variables (Supplementary Fig. 8)."

2. I am also still not clear on how to think about the difference between the choice signals in the pre- vs post-movement period. The authors have added some description that these choice signals have differential localization- pre being global and post being localized to the ventral stream (area L). However, little is done to describe whether these signals in fact reflect the same information or different information- and what that might be.

In addition, there is some lack of clarity on the separation of these two-time windows in the discussion. For instance, effort is given in figure 3 to demonstrate that there is choice information preceding the movement, and this is used to argue that this is therefore distinct from the left/right movement per se (lines 337-338). But this only applies to the pre-choice window. It is also notable that the pre-window is relatively weak (low d') while the post window is much stronger. More needs to be done to address the conceptual difference between these choice signals, and why they are orthogonal.

We thank the referee for requesting further clarification on this important point which was not sufficiently examined in the previous manuscript. We have now expanded our analyses and were able to provide new insights regarding the differences between the choice signals and what information they might reflect. Regarding “same/different information”, the choice axis both in the pre- and post-movement conditions is, by definition, the “maximally informative” linear discriminant of the animal’s choice. As the referee correctly pointed out, our previous analyses suggested that the ventral component was key to understanding the pre-post differences. The new Supp. Fig. 14a shows that the largest increase in discriminability from the pre- to the post-movement periods appears indeed in the ventral components. The d' increase from the ventral components is significantly higher than the increase associated with components from the dorsal and posterior areas. This result by itself could already explain the change in angle between the two axes (since a non-uniform change in the weights of a few components could result in orthogonal vectors). However, we also show (Supp. Fig. 14b) that even in the absence of ventral components (i.e., defining the choice axis using only components from non-ventral areas), pre- and post-movement axes are still orthogonal to each other. This result suggests that a “representational change” of choice information is broadly distributed in the posterior cortex and not restricted to ventral regions. However, it is in ventral regions that choice discriminability increases the most. A new paragraph has been added in the results section (Ln 237):

“These results highlighted the major differences between choice state axes defined pre- and post-movement. Choice pre-movement was less localized (lower SDI), with each cortical region contributing similarly. On the other hand, choice post-movement was more localized (higher SDI), with larger contributions in ventral stream areas (Fig. 4e). We also computed the increase in choice discriminability from the pre- to the post-movement periods and found that the d' increase from the ventral components ranked significantly higher than the increase associated with components from the dorsal and posterior areas (95% CI) (Supplementary Fig. 14a). Even in the absence of ventral components (i.e., defining the choice axis using only components from non-ventral areas), pre- and post-movement axes were still orthogonal to each other (Supplementary Fig. 14b, Discussion).”

What, mechanistically, causes this representational change is hard to pinpoint from our widefield data. We speculate it could be linked to a change in afferent choice

signals from pre- to the post-movement periods. For example, before the movement, choice signals could represent a distributed afferent from midbrain regions (Steinmetz et al., Nature, 2019) and post-movement choice signals might reflect “additional” feedback information from motor cortices where choice information is also encoded (Papel and Siegel, Nat Comms, 2016; Siegel et al., Science 2015), possibly with a stronger signature along ventral regions. Future studies examining brain-wide the functional feedback to these regions in similar DM tasks might shed light on these mechanisms. A new discussion regarding these considerations has now been added to the manuscript (Ln 370):

“We also verified that ventral stream responses were not linked to eye movements (Supplementary Fig. 15a), which typically followed whole-body movements²⁰, or to stimulus movement (Supplementary Fig. 15b) However, signals detected in ventral areas may still be associated with motor-related components that also carry choice-relevant information⁶⁷. The fact that pre- and post-movement choice axes remained orthogonal to each other in the absence of ventral components (Supplementary Fig. 14b) suggests a “representational change” of choice information broadly distributed in the posterior cortex and not restricted to ventral regions. What, mechanistically, could cause this change is difficult to establish solely from our widefield data. It could reflect a change in afferent choice signals from the pre- to the post-movement periods: for example, before the movement, choice signals could reflect a distributed afferent from midbrain regions²⁴, while post-movement, choice-related components from motor cortices^{67,68} could add to those from the midbrain and possibly with a stronger signature along ventral regions. Future studies examining functional feedback to these regions in similar decision-making tasks might shed light on these mechanisms.”

3. The authors have added trial numbers which are extremely helpful- and seem to be well above the threshold for significance. However, only a single number was given per A/B condition- and while some A/B conditions should be matched (e.g. sustained attention and stimulus), others are likely not (e.g. wheel movement and saccade) and so should have two numbers.

We have added an additional supplementary table reporting the number of trials used for the definition of each state axis (Supplementary Table 2):

Supplementary Table 2 | Trial conditions and number of trials used to obtain the different state axis used throughout the text.

State axis definition (maximize discriminability between trial sets A and B)	Trial set A (avg num of trials per animal)	Trial set B (avg num of trials per animal)	Notes
Stimulus	Stimulus present (2730)	No stimulus present (2730)	Number of no stimulus conditions chosen to match the number of stimulus (randomly chosen)
Wheel movements	Trials with a wheel movement (left or right) after the stimulus, without any other movements before the stimulus and without any saccades within 1s of the movement (1190)	Trials without a wheel movement (left or right) after the stimulus, without any other movements before the stimulus and without any saccades within 1s of the movement (1190)	Number of no wheel conditions chosen to match the number of stimulus (randomly chosen)
Saccades	Trials with a saccade after the stimulus, without any other saccades before the	Trials without a saccade after the stimulus, without any other saccades before	Number of no saccade conditions chosen to match the number of

	stimulus and without any wheel movements within 1s of the movement (465)	the stimulus and without any wheel movements within 1s of the movement (465)	stimulus (randomly chosen)
Sustained attention	Top 33 rd percentile of pupil area change trials (896)	Bottom 33 rd percentile of pupil area change trials (896).	
Choice	Trials with a right choice during the closed loop (657)	Trials with a left choice during the closed loop (721)	Pre- and post- movement axes were defined at different time points within the trial (before and after movement onset)
Attention on Corr. Trials	Top 33 rd percentile of pupil area change trials that resulted in correct choices (668)	Bottom 33 rd percentile of pupil area change trials that resulted in correct choices (592)	
Attention on Incor. Trials	Top 33 rd percentile of pupil area change trials that resulted in incorrect choices (221)	Bottom 33 rd percentile of pupil area change trials that resulted in incorrect choices (248)	
Choice with high attention and easy trials	Trials where the animal made a right choice during the closed loop on the top 33 rd percentile of pupil area change and with angle difference > 45 deg (142)	Trials where the animal made a left choice during the closed loop on the top 33 rd percentile of pupil area change and with angle difference > 45 deg (127)	
Difficulty with high attention and hard trials	Trials where the animal made a right choice during the closed loop on the top 33 rd percentile of pupil area change and with angle difference < 45 deg (177)	Trials where the animal made a left choice during the closed loop on the top 33 rd percentile of pupil area change and with angle difference < 45 deg (206)	
Contralateral stimulus information	Left stimulus horizontal, angle difference = 90 deg and left choice (70)	Left stimulus vertical, angle difference = 90 deg and left choice (70)	

REVIEWER COMMENTS

Reviewer #2 (Remarks to the Author):

The authors have addressed all of my concerns.